# Enhanced resolution capability of SWOT sea surface height measurements and its application in monitoring ocean dynamics variability

**Yong Wang[1]· Shengjun Zhang[1] \*·Yongjun Jia[2]**

1. School of Resources and Civil Engineering, Northeastern University, Shenyang, China.

2. National Satellite Ocean Application Service (NSOAS), Beijing 100081, China

Correspondence to: Shengjun Zhang (zhangshengjun@whu.edu.cn)

**Abstract.**
The wavenumber spectrum of sea surface height along ground profiles is commonly determined to
quantify the magnitude of detectable ocean dynamic features by altimetry missions. In this paper,
wavenumber spectral were calculated and compared for HY2B, Saral/AltiKa, Sentinel-3A, and SWOT.
The wavenumber power spectral density(PSD) of sea surface height (SSH) was averaged using weighted
methods across multiple along tracks within defined boxes. The deduced resolution capabilities were
also compared and analyzed, evaluated using the relevant definition of one-dimensional mesoscale
resolution capability. We verified that the latest wide-swath SWOT mission offers significantly improved
measurements. For example, in the vicinity of Kuroshio, the one-dimensional mesoscale resolution of
SWOT is about 25 kilometers, twice the resolution capability of conventional satellites. In addition, the
quality of measurements declined obviously over regions where the eddy kinetic energy gets larger.
Finally, the scale of change in global ocean dynamics between 60°N and 60°S was analysed using cross
power spectrum analysis based on SWOT data from two 21-day cycles. The results showed significant
geographic and temporal variations in the ocean dynamics variability scales, which are mainly relative
to sea state variability. The regions with large scales of ocean dynamics variability are concentrated in
oceans with strong currents and unstable sea states, such as the Kuroshio Current, the Gulf Stream, and
the Antarctic Circumpolar Current. In addition, the scale of ocean dynamics variability is not necessarily
large where eddy kinetic energy is large, such as the equator and the northwest Indian Ocean current area.
Ocean dynamics variability also varies across seasons. Meanwhile, SWOT one-day repeat cycle data
were also utilised to analyse the sub-mesoscale variability of the ocean.
# 1. Introduction
In recent decades, a series of satellites with altimeters onboard were launched, enabling continuous
monitoring of sea surface height (SSH) information on a global scale. These developments also led to a
better understanding of multi-scale dynamical phenomena (e.g., El Niño, Rossby waves, mesoscale
eddies, etc.) in the ocean surface (Boas et al., 2022). Within this, mesoscale dynamics are connected
through interactions with large-scale oceans (Smith et al., 2001). At the same time, mesoscale eddies
generate finer mesoscale and sub-mesoscale motions through small-scale frontal formations at the sea
surface (Lapeyre et al., 2008). Driven by different mechanisms of quasi-geostrophic (QG) dynamics,
sub-mesoscale activity can also reverse the cascade of energy from the sub-mesoscale to the mesoscale
energy (Cao et al., 2021; Qiu et al.,2022). Processes at these spatial scales, through kinetic energy
cascades and energy dissipation, are essential for determining the upper ocean energy transfer.
(McWilliams et al., 2016; Rocha et al., 2016).
Multi-satellite merged products have been widely applied in oceanography, such as detecting and
tracking mesoscale eddies(Samelson et al., 2014; Yi et al., 2014; Chen et al.,2019). However, the optimal
interpolation algorithm heavily smoothed the spatial scales below 200 km during the product
manufacturing process, preserving only a limited portion of the small-wavelength signals (Dufau et al.,
2016). Therefore, those merged products are only suitable for observing mesoscale ocean dynamics at
wavelengths exceeding 150-200 km. (Vergara et al., 2023; Samelson et al., 2014; Boas et al., 2022; Dufau
et al., 2016).It is impractical to use traditional satellite altimetry missions to study the two-dimensional
sub-mesoscale dynamics of the ocean. The Surface Water and Ocean Topography (SWOT) satellite,
successfully launched by NASA in December 2022, observes SSH through a 50-km strip on either side
of the satellite nadir. It is expected to be able to resolve two-dimensional SSH variability structures at
wavelengths down to 15 km (Boas et al., 2022; Chelton et al., 2019). This will also dramatically enhance
our understanding of upper ocean dynamical processes in the mesoscale to sub-mesoscale wavelength
range (15-200km).

46       Before SWOT, sub-mesoscale dynamics were unresolved by AVISO's merged products or by most

global eddy-resolved models. However, they can be partially captured by SSH observations along
satellite profiles. Along-track altimeter data offers higher spatial resolution than merged data. It is capable
of sustained, repeated sampling of the global oceans so that the variability of SSH can be analyzed and
statistically assessed. In particular, the estimation of the power spectral density(PSD) of its SSH
wavenumber can be used to analyze the energy and cascade(Dufau et al., 2016; Le Traon et al., 2008; Xu
and Fu, 2011, 2012). Based on the hypothesis of geostrophic balance, energy conservation, and potential
vortex conservation, the wavenumber spectrum of SSH lies between the quasi-geostrophic (QG)
turbulence theory and the surface quasi-geostrophic (SQG) turbulence theory (Xu and Fu, 2012; Qiu et
al., 2018; Chereskin et al., 2019; Callies et al., 2011). The theory predicts that the spectral slope of the
wavenumber (K) varies from $k^{-5}$ to $k^{-\frac{11}{3}}$ in the mesoscale to sub-mesoscale scale range. Xu and Fu.
(2012) first utilized Jason1 data and accounted for the effect of noise to perform chunk statistics on the
PSD of global SSH wavenumber. They found that in the energy core regions of major ocean currents
(e.g., Kuroshio, Gulf Stream, Antarctic Circumpolar Current, Brazilian Warm Current, etc.), their slopes
can be observed to be between the QG turbulence theory and the SQG turbulence theory. Besides these
high kinetic energy regions, the slopes in the temperate and tropical zones are significantly lower than
$k^{-\frac{11}{3}}$, which they attribute mainly to the influence of non-geostrophic dynamics.

63       Dufau et al.(2016) proposed a method defined as the one-dimensional mesoscale resolution

capability of altimetry satellites. The slope is determined by fitting the 90-280 km wavenumber spectrum
and using 25 km below as the noise constant. Notably, diverse methods for calculating the Power Spectral
Density (PSD) can result in minor discrepancies in the slope range(Vergara et al., 2019). Additionally,
data sampled at varying frequencies may also engender subtle variances in the estimated PSD slope range.
The intersection between these two is defined as the one-dimensional mesoscale resolution capability.
There are significant differences in resolution capability attributable to varying noise levels of altimeters
in different frequency bands and modes. The Jason2 uses Ku-band low-resolution mode (LRM), which
typically only resolves wavelengths of about 70 km (Vergara et al., 2019). The Saral/AltiKa uses a 40
HZ Ka-band transmitting frequency, a wider bandwidth, and a higher pulse repetition frequency, resulting
in lower noise levels than Ku-band LRM altimeters(Raynal et al., 2018). Thus, it exhibits a higher one-
dimensional mesoscale resolution capability than the Jason satellite (Dufau et al., 2016). The altimeter
of Sentinel-3A uses the Ku-band synthetic aperture radar (SAR) mode, which achieves lower noise
compared to Jason2 and Saral/AltiKa. (Vergara et al., 2019). Its true resolution capability is also better
than Jason2's LRM mode and the Saral/AltiKa satellite (Raynal et al., 2018). The latest SWOT satellite
uses a Ka-band radar interferometer (KaRIN), increasing spatial resolution along the track to 2 km and
thereby greatly enhancing our understanding of sub-mesoscale dynamics (Callies et al., 2019).
Accordingly, this paper presents an updated analysis of the global SSH spectral slope between 60°N and
60°S, using data from various altimetry missions. The method of Dufau et al. (2016) was adopted and
improved to statistically analyze the one-dimensional resolution capability of altimetry satellites with
different modes and frequencies to further validate the enhancement brought by SWOT satellites. Finally,
the dynamical scales of the worldwide between 60°N and 60°S ocean are analyzed through SWOT data.

This paper is structured as follows. In section 2, we describe the altimetry dataset used and a specific description of the method improvement. In section 3, we statistically assess the noise levels of altimetry satellites of different modes and frequencies and the worldwide between 60°N and 60°S one-dimensional mesoscale resolution capability. Section 4 defines a parameter using cross power spectral analysis and analyzes worldwide between 60°N and 60°S ocean dynamics variability at the mesoscale and sub-mesoscale using SWOT data. Finally, we summarize the enhancement brought about by the SWOT satellite.

# 2. Datasets and Methodology

## 2.1 SSH Datasets

Along-track SSH data from four altimetry missions (HY2B, Saral/AltiKa, S3A, SWOT) using different techniques are analyzed on a global ocean scale. SWOT provides two-dimensional sea surface height observations. We subsample the original two-dimensional gridded data and split it into 69 separate one-dimensional along-track data (That is, the two-dimensional data has 69 cross-track points), and selected the 15th and 45th of these as experimental data. For Section 3, we only selected data from October 2023 for analysis due to the large amount of SWOT data. The resolution capabilities of different altimetry techniques are compared to validate the higher resolution enhancements brought by SWOT's KaRIn. For Section 4, data from 8 cycles of the SWOT mission are used. For the first three missions, we select only 1hz data; for SWOT data, we choose the cross-corrected oceanic Level 3 product. The details of the four missions are described below.

The HY2B satellite mission Level 2 products are all released by the National Satellite Ocean Application Service Center of China (NSOAS, https://osdds.nsoas.org.cn/home), with a repeat cycle of 14 days. The satellite mainly carries dual-frequency radar altimeter (Ku and C bands), and the Ku band is mainly used for distance measurement. HY-2B satellite radar altimeter secondary products include Operational Geophysical Data Records (OGDR), Interim Geophysical Data Records (IGDR), Sensor Geophysical Data Records (SGDR), and Geophysical Data Records (GDR). IGDR is an uncorrected data product obtained using Medium Orbit Ephemeride (MOE) orbit data, waveform reconstruction, etc. GDR is a fully corrected data product obtained using Precise orbit ephemeris (POE) orbit data, waveform reconstruction, etc. SGDR is the same as GDR, but the difference lies in including waveform data. In this paper, we use SGDR data for HY2B, with a time horizon of October 2023.

The SARAL/AltiKa satellite was launched as a collaboration between the Indian Space Research Organisation (ISRO) and the French National Centre for Space Studies (CNES) (Verron et al., 2015), with a repetitive period of 35 days. Using the Ka-band allows for reduced size, lower ionospheric attenuation delays, and higher measurement accuracy than conventional Ku-band altimeters (Quartly et al., 2015). In ocean observations, it improves the accuracy of SSH, especially for ocean mesoscale observations (Verron et al., 2021). The advantages of the Ka-band are reduced ionospheric effects, smaller footprint, better horizontal resolution, and higher vertical resolution (Verron et al., 2015; Smith et al., 2015). A disadvantage of the Ka-band is the attenuation in rainy conditions due to water or vapor and the resultant loss of data (Lillibridge et al., 2014). Finally, SARAL/AltiKa data with a period of October 2023 was selected.

The Sentinel-3A (S3A) satellite carries the Synthetic Aperture Radar Altimeter (SRAL) for distance measurements, which is processed using delayed Doppler processing designed to achieve significantly higher signal-to-noise ratios (Heslop et al., 2017). The main frequency used for distance measurements

is Ku-band (13.575 GHz with a bandwidth of 350 MHz) and C-band frequency (5.41 GHz with a bandwidth of 320 MHz) is used for ionospheric correction. There are two radar modes, Low Resolution Mode (LRM) and Synthetic Aperture Radar (SAR) mode. The SRAL mission on S3A always operates in high-resolution mode (often referred to as SAR mode). The repetition period of the S3A sun-synchronous orbit was 27 days. The data for S3A in October 2023 was selected.

The SWOT provides the first two-dimensional high-resolution measurement of water height from space using two SAR antennas separated by a 10-meter mast for interferometry in orbit. SWOT adopts Ka-band radar interferometry (KaRIn) for measurements over a narrow strip of 120 kilometers (a nadir 20-km gap is supplementally measured by a conventional altimeter at a low resolution). SWOT carries a Ka-band radar interferometer with azimuthal resolution of 2.5 m and distance resolution of 10-70 m. The pixel sizes of a few tens of meters are much smaller than the pulse-limited footprint area (~10 km) of conventional altimeters, and the high resolution of the radar system permits averaging over a large number of pixels to minimize noise and still resolve small-scale signals (Fu et al., 2024). In this paper, SWOT's latest cross-corrected L3 product is used. Data from SWOT's ocean product for cycles 1 through 14 are selected for this paper. Data for the latter three missions are available for download from the AVISO website (https://www.aviso.altimetry.fr).

The along-track SSH (HY2B, SARAL/AltiKa, S3A) observations were kept at their original 1hz observation positions at intervals of about 7km and corrected for environmental, and geophysical corrections. The corrections of the HY2B satellite are mainly made according to the data editing guidelines given in the satellite user's manual, and they are corrected during the satellite data pre-processing. The correction kinds mainly include dry troposphere correction, wet troposphere correction, ionosphere correction, sea state correction, ocean tide correction, solid earth tide correction, polar tide correction, and atmospheric inverse pressure correction. The other two satellites, SARAL/AltiKa and S3A, were directly fed through AVISO's Level 2 sea surface height anomaly product. Both products undergo a similar pre-processing step as the HY2B satellite. The SWOT mission selects data in the along-track direction with an interval of 2 km. All missions follow (Xu and Fu, 2011, 2012; Dufau et al., 2016) in calculating SSH anomalies by subtracting from the along-track SSH measurements the SSH anomalous wavenumber Mean Sea Surface Model (MSS) CNES_CLS_2015, which is the time-varying portion of the SSH.

## 2.2 Methodology

Wavenumber spectral analysis is a common method to study the characteristics of a signal or system within the frequency domain. The spectral signal is obtained by sampling the signal in the time domain and performing a Fourier transform on it, then sampling it in the frequency domain to obtain a frequency domain signal. Wavenumber represents the number of wavelengths per unit distance. The wavenumber spectrum indirectly reflects the energy distribution in the ocean at different spatial scales. The shape and characteristics of the spectrum can provide important information about the underlying physical processes(He et al., 2024).

Although we want to obtain results with a resolution of 2° x 2° between 60°N and 60°S globally, each grid point will be expanded into a 10° x 10° box for calculating statistics. We performed a Fourier transform on the along-track SSH anomaly data for each mission in a box and calculated its wavenumber power spectral density (PSD). The specific preprocessing steps for calculating the PSD for each along-track SSH within a box are similar to those described by Dufau et al.(2016). For each 10° × 10° box, instead of simply assigning the same weighting to all individual PSDs within each box to calculate the

average, we propose a new approach based on trajectory distance weighting to calculate the average PSD for each box. This is because averaging over a $10° \times 10°$ area would diminish the signal in regions with higher mesoscale energies, as well as affect the assessment of areas with lower energies., leading to larger errors in the results. Consequently, we follow the method in Appendix A to calculate the distance between each SSH along the track and the reference point. Then, the weight of the PSD for each SSH in the region is assigned based on this distance. The method of weighted averaging can reduce the error in calculating the PSD of the grid points and preserve the signal of the grid point location as much as possible so that the calculation results can be more credible. Finally, using the PSD after weighted averaging, three parameters were derived using the method of Dufau et al. (2016): the 1 Hz SSH error level (SWOT is the 2 km sampling resolution along the track), the SSH spectral slope in the mesoscale bands, and their intersections, expressed as wavelengths, referred to as the "one-dimensional mesoscale resolution capability".

For wavelengths below 25 km for the first three missions, the 1 Hz SSH error level was estimated by fitting a level to the spectrally flat noise levels present in the PSD maps (Figure 1). SWOT error levels were estimated by fitting a level to noise levels below 15 km, adopting the results of Chelton et al. (2019) on the assessment of noise levels in SWOT. These arise primarily from in homogeneities in the radar backscatter coefficient within the altimeter footprint, resulting in inaccuracies in SSH estimates and generating greater spectral noise.

The PSD of SSH is estimated by first removing the estimated constant error level to allow for an unbiased estimate of the spectral slope. For the first three conventional missions, we chose wavelengths in the range of 70-250 km and fitted the slope of the PSD by least squares (Le Traon et al., 2008; Xu and Fu, 2011; Dufau et al., 2016;). The lower limit was chosen to ensure a robust slope estimate, as the shape of the PSD exhibits greater variability below this limit (e.g., Figure 1). The 70-km wavelength exceeds the shorter wavelengths affected by altimeter noise. For the SWOT mission, the cross-correction process filters out some of the noise, which results in a continuous decrease in the spectral profile at less than 15 km (as shown in Figure 1), in addition to many sub-mesoscale phenomena such as internal waves and tides at 15-40 km (Boas et al., 2022). Therefore, a wavelength range of 40-125 km was selected for calculating the PSD slope for the SWOT mission. The intersection points where the error level and the spectral slope define the wavelength at which the PSD of the smallest-scale signal equals the error level. It is also called the satellite's one-dimensional mesoscale resolution capability. We will also use this parameter to compare the resolution capability of different modes of altimetry satellite missions in Section 3 to assess the improvement in mesoscale resolution capability provided by the SWOT satellite.

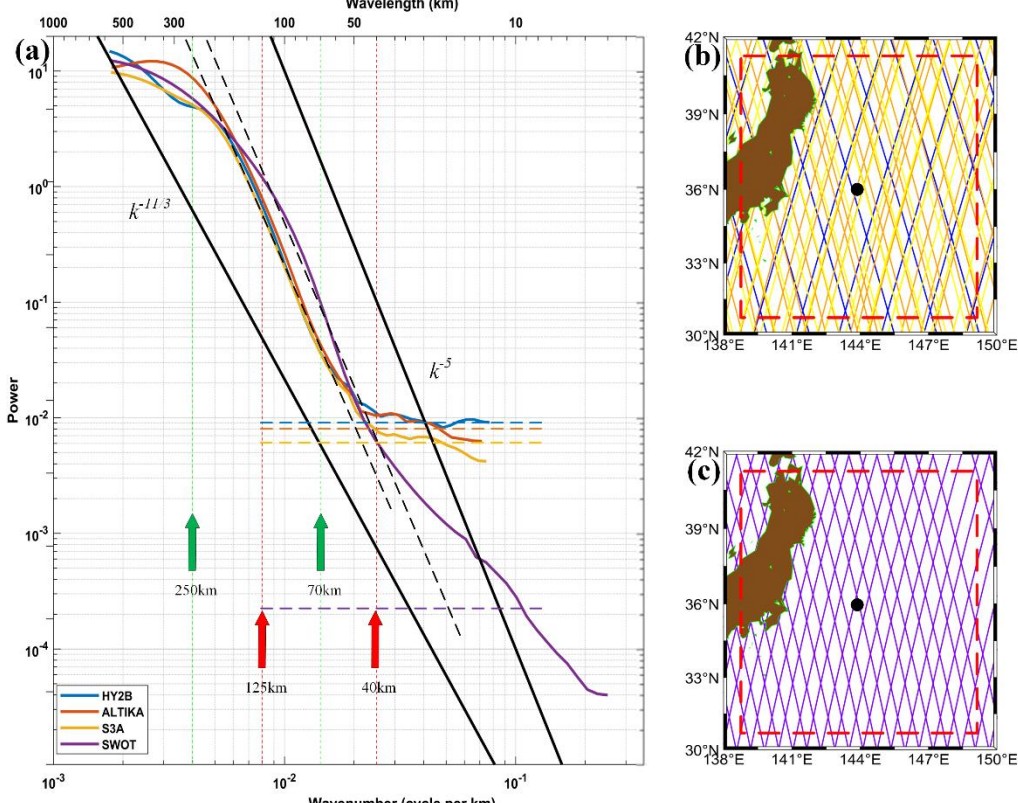

Figure 1 a. Along-track SSH averaged PSDs for HY2B, SARAL/ALTIKA, S3A, and SWOT within the Kuroshio Extension (all spectra in Figure 1a are biased, green arrows represent the range over which PSD slopes were computed for conventional satellites, red arrows represent the range over which PSD slopes were computed for SWOT satellites, and the black solid lines show the spectral slopes which correspond to $k^{-5}$ and $k^{-11/3}$). b. The tracks of the first three satellites within the Kuroshio region distribution map. c. Distribution map of SWOT satellite tracks within the Kuroshio region (only two along-track data were selected for each pass). Red dashed lines represent the range over which the mean PSD was computed. The horizontal four dashed lines represent the constant noise level for each satellite, while the two black dashed lines represent straight lines fitted to the unbiased spectra after removing constant noise from the HY2B and SWOT spectra, respectively.

# 3. Resolution capability of altimetry satellites

To compare the difference between the new method of calculating the slope and the previous method, we used the HY2B satellite and calculated the global SSH spectrograms for both methods (Figure 2). We can observe from Figures 2a,2c. When counting the global distribution maps, distance-weighted averaging can reflect the spatial correlation of geographic phenomena more accurately by assigning distance-based weights to the observation points. This method not only reduces the bias caused by local outliers or uneven data distribution, but also enhances the regional representativeness of the distribution map. Figure 2b,2d shows the zoomed-in local area, and it can be clearly observed that the distance-weighted method can more accurately bring out the detailed part of the SSH slope map. In contrast, the traditional equal-weighted averaging method assigns the same weight to all observations, ignoring the

effect of distance on the statistical results. This approach tends to lead to excessive smoothing of the signal, especially when analysing subtle changes such as slope, and may mask important local features, thus reducing the accuracy and explanatory power of the distribution plot. Therefore, the use of the distance between trajectories to adjust the weights improves the accuracy and reasonableness of the global distribution statistics and provides a more reliable basis for subsequent analyses.

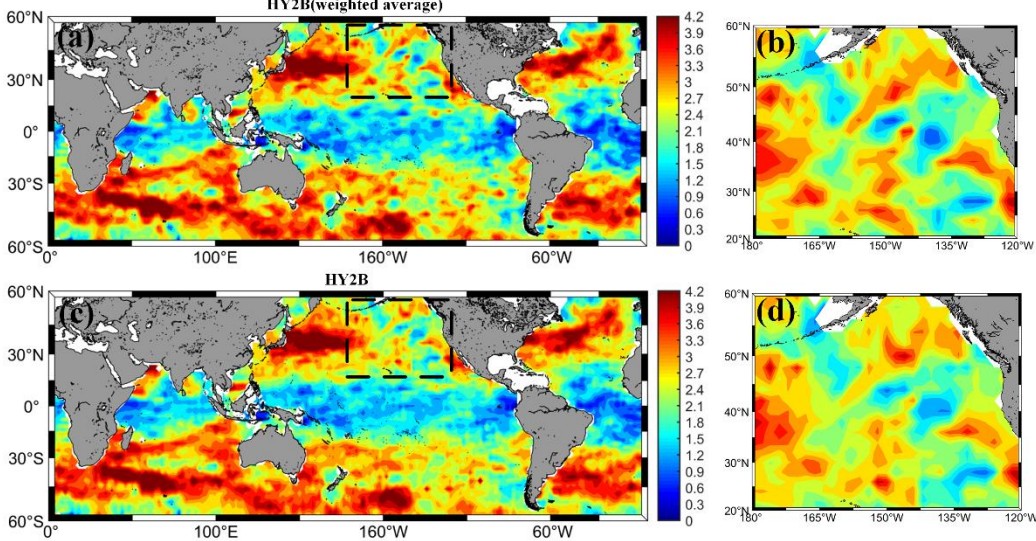

**Figure 2 Distribution plots of weighted versus equal weighted averaging (a. Results of weighted averaging using distances between satellite tracks. b. The plot of results averaged using the same weights. c and d are areas enlarged by the black boxes in a and b, respectively. Units=log (m$^2$/cpkm )/log(cpkm))**

The worldwide between 60°N~60°S maps of SSH spectral slopes from October to November 2023 for the four different model altimetry missions are almost identical (Figure 3). They are in general agreement with the map of slopes averaged over longer periods by Dufau et al.(2016) and Xu and Fu.(2012). The highest slope distributions for each altimetry mission were observed in major ocean current regions, including the Kuroshio, Gulf Stream, Antarctic Circumpolar Current, Brazilian Warm Current, and the North Indian Ocean Current. The spectral slopes in these regions are all close to $k^{-4.2}$ or even higher, which is consistent with the theoretical predictions from SQG and QG theories(Xu and Fu,2011). The slopes are lower at lower latitudes, typically below $k^{-2.1}$. Figure 3 in the Dufau et al. (2016) article pointed out that the calculated PSD shows an important energy peak near the 140 km wavelength at low latitudes. These peaks correspond to residual tidal signals affecting the altimeter's SSH measurements, which are of varying strengths but may be hidden at mid-latitudes by the higher geostrophic energy occurring at similar wavelengths. These peaks may be related to errors in the positive pressure tidal correction. This may be due to uncorrected baroclinic tides in the altimeter's SSH measurements (Richman et al.2012). Another explanation is that the geostrophic equilibrium motions (i.e., mesoscale eddies) in these regions are lower than those of Kinetic Energy (KE) levels (i.e., internal waves). Thus, the latter would mask the energy levels associated with mesoscale eddies (Tchilibou et al.,2018). This explains why satellite altimetry observes relatively small spectral slopes at these latitudes. Vergara et al. (2019) combined spectral slopes with local stratification and Rossby number, and used variable wavelength ranges to fit the slopes of the spectra. However, for this paper, we focus on verifying the improvement brought by SWOT compared to other mission satellites. Therefore, for the one-

dimensional mesoscale resolution capability of different altimetry missions, we compare and analyze only the mid-latitude regions where energy levels are higher.

We observe that the spectral slopes of SWOT are consistently higher than the results of the other three missions. This may be related to the fact that the noise levels of HY2B, ALTIKA, and S3A hide the ocean variability at a wavelength of about 70 km and will result in a smaller slope. It is also possible that slight differences in the slopes arise from varying sampling rates, but the map's distribution pattern is essentially consistent. Although our calculated spectral slopes for one month are similar to the results of previous studies (Xu and Fu,2011; Dufau et al.,2016). However, there are still slight differences in the results due to differences in the method of calculation, data processing, and the period of the study. In fact, several studies have demonstrated that SSH spectral slopes exhibit seasonal variability. For example, in regions such as the Kuroshio, the Gulf Stream, and the northwest Pacific, the spectral slopes are stronger in summer and fall and weaker in spring versus winter (Dufau et al.,2016; Vergara et al., 2019). The fact that our data were chosen in October may explain why the spectral slopes calculated in this paper are slightly larger.

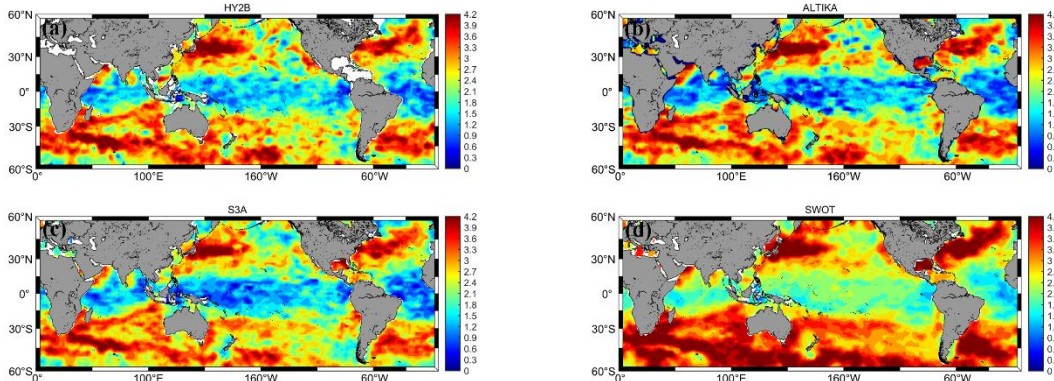

**Figure 3 Slope maps for different satellites ((a) HY2B, (b) SARAL/ALTIKA, (c) S3A, (d) SWOT. Units=log(m$^2$/cpkm)/log(cpkm))**

Figure 4 depicts the spatial distribution of altimeter error levels over the worldwide between 60°N~60°S ocean for the four altimetry missions HY2B, SARAL/ALTIKA, S3A, and SWOT. As can be seen from Figure 4, HY2B uses the highest noise LRM mode in the Ku-band. The Ka-band used by SARAL/Altika demonstrates a significant reduction in the altimeter noise level. However, during the study period of this paper, the SARAL/Altika satellite's trajectory data in specific regions show significant deviations with a wide range of effects (see Figure .4b). Through a detailed examination of the correction terms and comparative analysis of data with the same orbit number in two adjacent cycles, we find that the main source of this trajectory effect is the problem of satellite orbiting accuracy. The inaccuracy of the satellite orbiting leads to a significant increase in the data noise level, which results in a clear trajectory effect in the error level map. Additionally, the error of the Ku-band SAR mode altimeter on the S3A satellite has shown significant improvement over the previous two modes. The error magnitude of the Ka-band interferometric mode on the latest SWOT satellite has been reduced by over 70 times compared to traditional satellites. Comparing Figure 4 with Figure 5 reveals that, regardless of the altimetry mission type, all exhibit similar distribution patterns, with higher altimeter noise in regions of greater eddy kinetic energy. This results in an increased level of altimeter error below 25km (15km). It is necessary to upgrade the technology in the future or use appropriate algorithms to minimize the effect of this error.

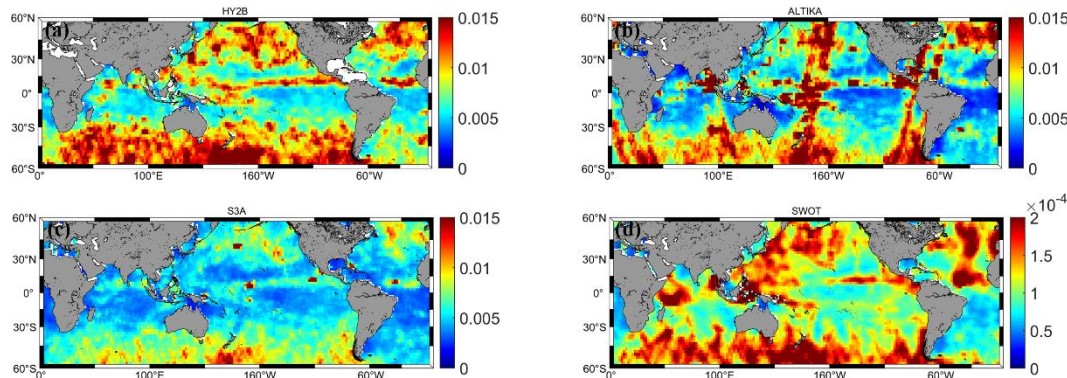

**Figure 4 Noise levels (m rms) of different satellites ((a) HY2B, (b) SARAL/ALTIKA, (c) S3A, (d) SWOT)**

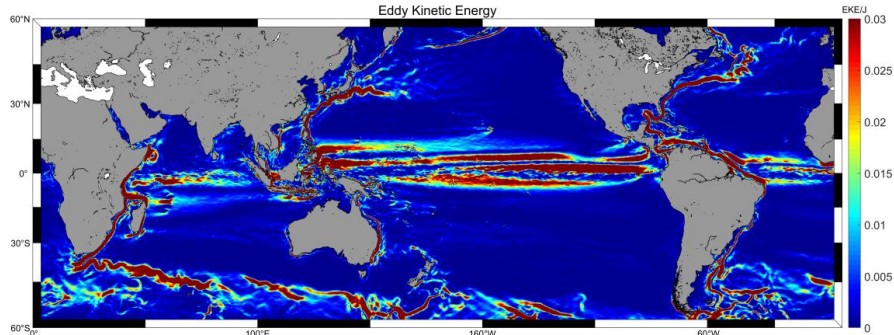

**Figure 5 Distribution of eddy kinetic energy calculated by MDT2022**

Based on the one-dimensional mesoscale resolution capability parameter defined by Dufau et al. (2016), we plot the distribution of the worldwide between 60°N~60°S observed minimum wavelengths for the four altimetry missions (Figure 6). It is evident that the resolution wavelengths calculated for different altimetry missions and geographic locations are different. This variation is primarily due to differences in noise levels and PSD slopes. There may be some issues with the calculated slopes at low latitudes, so we compare the main ocean current region (Kuroshio). The resolution capability of the LRM mode in the Ku band of HY2B is relatively poor, around 60 km. t The SARAL/ALTIKA mission achieves resolution at wavelengths greater than 50 km. In contrast, the SAR mode of S3A achieves resolution at wavelengths slightly below 50 km. This result is consistent with the conclusions reached by Vergara et al. (2019) and Dufau et al.(2016). Although they studied the Jason1 satellite, both Jason1 and HY2B satellites use the Ku-band LRM mode for their measurements. Finally, as shown in Figure 6d compared to Figures 6a, b, and c, the KaRIN approach adopted by SWOT represents a significant advancement in resolution capability. The Kuroshio waters can be resolved at wavelengths around 20 km. This is mainly due to SWOT's significantly lower noise level and the higher range of wavenumbers where the peak slope occurs (Figure1a). This paper only compares the along-track resolution capability of different missions. The primary advantage of SWOT lies in its capability to conduct two-dimensional SSH observations, which will provide unprecedented insights into small-scale ocean features. Experiments have demonstrated that SWOT's along-track resolution capability has also been greatly improved(Fu et al., 2024), providing a solid foundation for utilizing two-dimensional SSH data to study ocean dynamics.

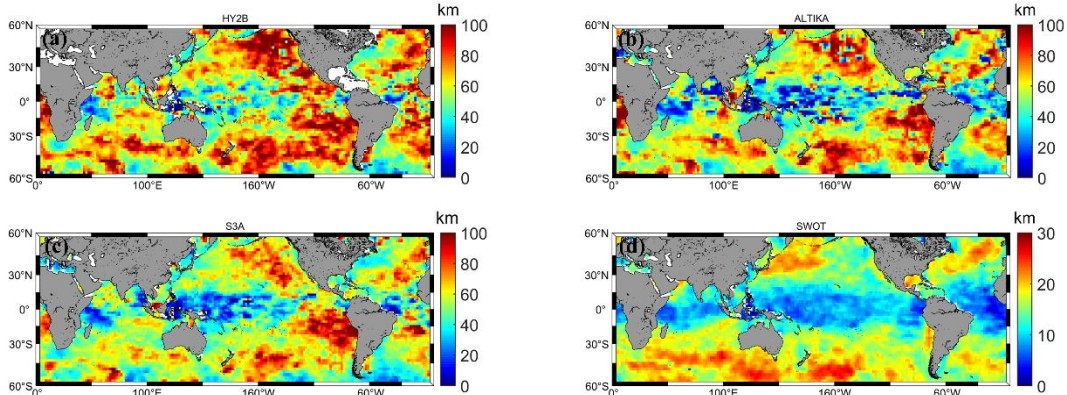

**Figure 6 Resolved wavelengths for different satellites (a. HY2B, b. SARAL/ALTIKA, c. S3A, d. SWOT)**

# 4. Analyses of ocean-scale changes

We confirmed in the previous section the great enhancement brought by SWOT. This section focuses on the study and analysis of the dynamical scales in the ocean using data from different cycles of SWOT. Eight cycles were selected (from four different seasons), with two cycles grouped together for each experimental set. The four groups are cycles 13 and 14 representing spring, cycles 1 and 2 representing summer, cycles 4 and 5 representing autumn, and cycles 9 and 10 representing winter. Each of these datasets is calculated using the Appendix B cross spectral density. Here, instead of utilizing the time-varying signal from the along-track SSH, we combine the ocean time-varying signal with the mean dynamical topography (MDT) to obtain the absolute dynamical topography (ADT). The specific subregional calculations are aligned with Appendix A.

In this paper, we use the method of Marks et al.(2016) to calculate the cross spectral density of data from the same geographic location for two adjacent cycles and the corresponding spatial wavelengths when the mean square coherence reaches 0.5. The two cycles are separated by 21 days, and we define this parameter as the wavelength of the Ocean Dynamics Scale (ODS) that is observed at that geographic location on day 21. A map of ODS variability in the global ocean between 60°N~60°S was constructed for four seasons according to the aforementioned standard (Figure 7). As can be seen from Figure 7, ODSs larger than 100 km are found mainly along the Western Boundary Currents and the Antarctic Circumpolar Current. These regions exhibit high variability in mesoscale ocean dynamics and unstable oceanic phenomena. Notably, regions with higher eddy kinetic energy do not necessarily exhibit greater ODS variability. For example, in equatorial regions and the northwest Indian Ocean current, both exhibit strong eddy kinetic energy (Figure 5). However, their ODS is relatively small during the period covered by the four experimental cycles, resulting in wavelengths that drive their variability being small, around 90 km.

There are different ODS variations in different geographic locations, and in the Western Boundary Currents, the Gulf Stream has greater oceanic-scale variability than the Kuroshio region. In addition, the ODS variations at the same geographic locations vary seasonally. For example, at the confluence of the Oyashio and Kuroshio, wavelength variation is greater in spring compared to winter and smaller in summer compared to autumn. In contrast, in the western location of Australia, there is less variation in spring and winter and more variation in summer and fall. Except for the ocean current regions, the rest of the areas show minimal variation in ODS, generally below 60 km. The experimental results in this paper do not indicate the absence of small-scale changes in the region but rather calculate the largest-

scale fluctuations in the area. The ODS of different ocean currents also differ, but it is important to note that the regional ODS of these ocean currents are generally greater than that of other locations. The parameter proposed in this paper can also be applied to SWOT data from one-day repeated cycles, which better represents ocean dynamics variability at temporal dynamic scales. It will also provide a novel reference point for future scientific research.

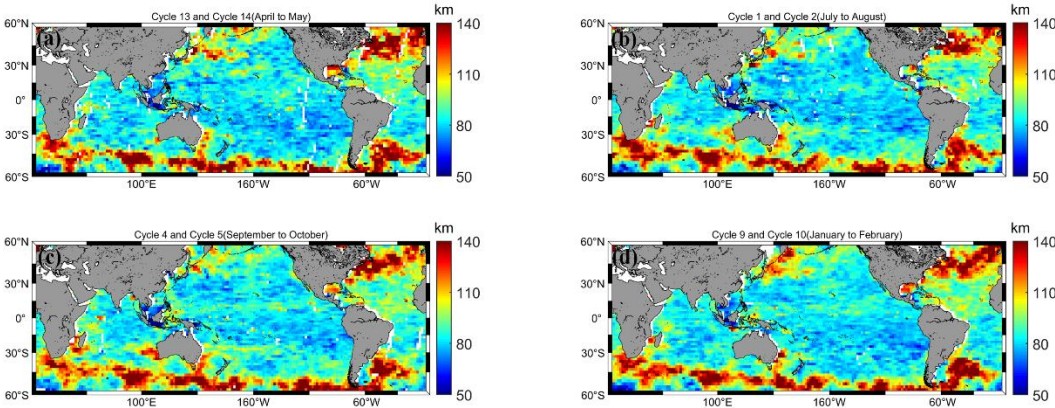

**Figure 7 Ocean mesoscale and sub-mesoscale scale changes over four seasons**

To better illustrate the role of SWOT one-day repetitive cycle data, we utilised two repetitive cycle trajectory data, SWOT_L3_LR_SSH_Expert_478_021_20230402T142629_20230402T151735_v1.0.nc and SWOT_L3_LR_SSH_ Expert_479_021_20230403T141706_20230403T150812_v1.0.nc. During the analysis, we divided the SWOT data into two strips along the track direction for the calculation. Of the multiple along-track data within each strip, we selected 100 data points as the data length for calculating the cross spectral density. The calculation results for each strip were obtained by averaging the data from multiple along-track directions. As shown in Figure 8, the cross spectral density computed using the same pass data on these two days reveal the scale differences in the ocean dynamics variations at different locations. Figures 8b, 8c and 8d, 8e show the magnified images of sea surface height anomalies at the same locations on 2 April 2023 and 3 April 2023 for regions 1 and 2 in Figure 8a, respectively. As can be seen from Figure 8a, region 2 experienced a scale change of about 30-35 km wavelength in just two days, while region 1 had a smaller scale change of about 20-25 km. This suggests that there are significant differences in the scale of ocean dynamics in different regions, even within the same time period. In addition, within the same region, the variability varies between strips, mainly because different regions are affected by different sub-mesoscale phenomena. Sub-mesoscale dynamics variability in the ocean is constantly occurring, which leads to differences between the left and right SWOT strips. However, this experiment only considered the direction along the satellite track and analysed the scale variation of ocean dynamics in the one-dimensional direction along the track. At the same time, the averaging of data from multiple along-track directions may diminish the significance of ocean dynamics scale variations within a local region. The aim is to provide a feasible framework for subsequent related studies and to lay the foundation for more in-depth ocean dynamics studies. Subsequent work will continue to be devoted to the study of ocean dynamics scale variability in two dimensions using SWOT.

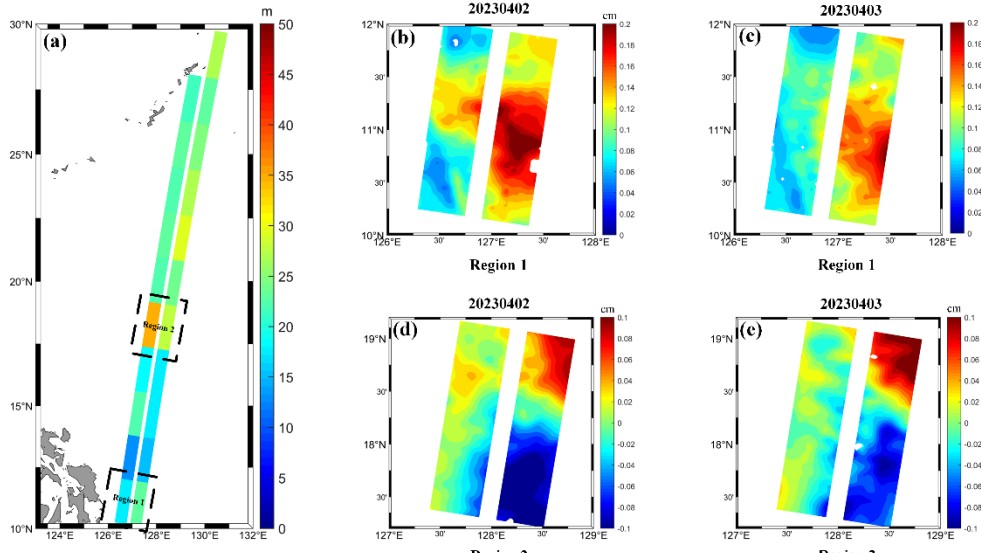

379
380 **Figure 8 Ocean dynamics scale changes calculated using SWOT one-day repeat cycle data**

## 5. Conclusion

We assessed the capability of four different modes of satellite altimetry (Ku-band LRM, Ka-band LRM, Ku-band SAR mode, and Ka-band wide mode) for global ocean between 60°N~60°S mesoscale resolution by comparing the PSD of their SSH along the track. In contrast to traditional averaging methods, this paper uses a weighted averaging method to calculate the mean value based on the distance between satellite orbits. This method provides a more accurate reflection of the noise level and resolution capability. The results show that the SWOT mission provides a significant improvement in along-track resolution capability compared to conventional one-dimensional altimetry satellites, especially in terms of noise level and one-dimensional mesoscale resolution. For example, in the vicinity of Kuroshio, the one-dimensional mesoscale resolution capability of SWOT is about 25km, which is about double the resolution capability of conventional satellites. In addition, we find that regions of high noise levels often correspond to regions of strong eddy kinetic energy. The higher the eddy kinetic energy, the relatively higher the noise level of the satellite.

Finally, by correlating eight cycles of SWOT data, we find significant ODS variations in the major ocean currents, including the Western Boundary Currents and the Antarctic Circumpolar Current. It is noteworthy that ODS variations are not significant in the warm equatorial current region despite its high eddy energy. In addition, seasonal ODS variations were also observed for major ocean currents such as the Kuroshio and Gulf Stream, with the Kuroshio showing larger ODS variations in the spring and winter, and smaller ones in the summer and autumn. Meanwhile, the sub-mesoscale variability of the ocean was analysed using SWOT data from two one-day repetitive cycles, revealing regional differences in ocean dynamics variability. This study not only demonstrates the improvement in the resolution capability of SWOT along-track data but also lays the foundation for future studies of oceanic sub-mesoscale dynamics using two-dimensional SSH data from SWOT.

## Data Availability Statement

SWOT data, Sentinel-3A altimeter data, and SARAL/AltiKa altimeter data were provided by CNES (https://www.aviso.altimetry.fr/en/data.html). The HY2B satellite mission Level 2 products are all released by the National Satellite Ocean Application Service Center of China (NSOAS,

http://www.nsoas.gov.cn/). The MDT2022 can be downloaded from the AVISO website
(https://www.aviso.altimetry.fr).
**Conflict of interest** The authors declare that they have no conflict of interest.
**Author contributions**
Wang Yong analyzed the data and wrote the manuscript draft; Zhang Shengjun and Jia Yongjun reviewed
and edited the manuscript.

# Appendix A: Detailed description of the PSD averaging method for multiple tracks within the box

To avoid the effect of sea ice on SSH, we divided the globe ($0°E \sim 360°E, 60°N \sim 60°S$) into 10,980
small $2° \times 2°$ regions. The PSD at each grid point was calculated by extending the area to a $10° \times 10°$
box. To enhance the accuracy of the PSD at each grid point, we first calculated the distance $D$ from the
SSH along the track to the grid point as follows.
The trajectories of the satellites in the $10° \times 10°$ box resemble a parabola, as shown in Figure A1.
Therefore, we model the trajectory of each satellite using a binomial equation, as illustrated in Eq 1.

$$y_i = ax_i^2 + bx_i + c \tag{1}$$

where $i$ represents the number of compliant along-track SSHs in the box. $x_i$ is the longitude
independent variable and $y_i$ is the latitude dependent variable.
Assuming that the position of the grid points to be solved is $(x_0, y_0)$, the distance $D_i$ can be expressed
by Eq. 2.

$$D_i = \sqrt{(X_i - x_0)^2 + (Y_i - y_0)^2} \tag{2}$$

where $(X_i, Y_i)$ is the position of the nearest point to the point to be sought in $i$ trajectories. Therefore,
we only need to find the position of the nearest point to find out the shortest distance from the point to
be sought to the trajectory $D_i$. The relationship equation for distance minimization is established by
combining Eq1 and Eq 2.

$$f(x_i) = \sqrt{(x_i - x_0)^2 + (ax_i^2 + bx_i + c - y_0)^2} \tag{3}$$

When the distance is minimum, the derivative of $f(x_i)$ should be zero at this point. Finally, it is
simplified to Eq 4.

$$x_i - x_0 + ax_i^2 + (bx_i + c - y_0)(2ax_i + b) = 0 \tag{4}$$

The solution $X_i$ of Eq. 4 needs to be computed using numerical iteration, and here we utilize
Newton's iterative method for the solution. Finally, the shortest distance $D_i$ can be found.

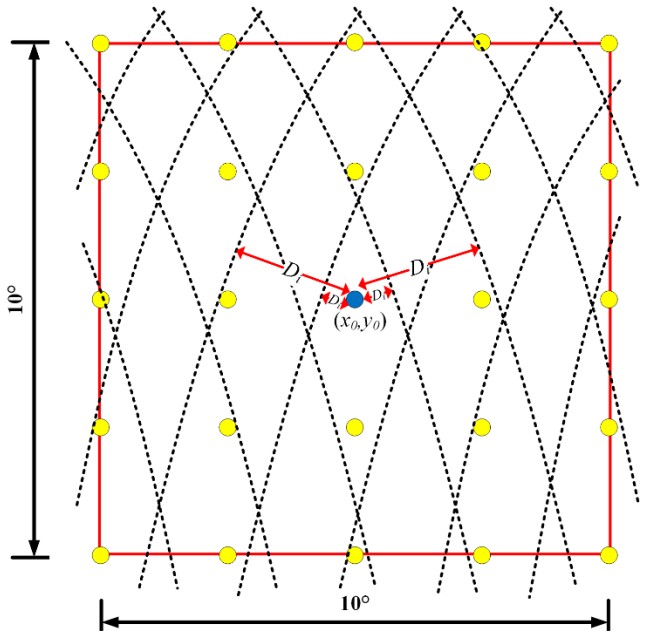


**Figure A1 Schematic diagram of the satellite trajectory and the grid points to be solved, where**
**the blue points are the grid points to be solved, the yellow points are the other grid points, the black**
**dashed line are the trajectories of the satellites, and the red lines represent the range of the box.**

After calculating the distances of all the trajectories inside the box to the grid points to be solved,
we calculate the average PSD of the grid points using the Inverse Distance Weighting (IDW) method.
Since we are calculating for the $2° \times 2°$ small areas. Thus, the weights within a $1°$ extension of the point
to be sought in all four directions should be the same. To facilitate the calculation, we define the weights
in the form of a circle expanding outward. The specific distance variations are shown in Figure A2. All
distances $D_i$ are rounded to the right, e.g., the green region distances are all $1°$. However, the distances
of the outermost black regions are all $5°$, which is due to the smaller weights they occupy, so they are set
to the same distance. Finally, we then weighted and averaged the PSD values of the points to be sought
according to Eq 5.
$$P = \frac{\sum_{i=1}^{n} w_i P_i}{\sum_{i=1}^{n} w_i}$$ (5)
where $w_i$ is $1/D_i$, $D_i$ is the PSD of the $i$ along-track SSH, and $n$ represents the total number of all
tracks in the box. P is the average PSD value of the point to be sought.

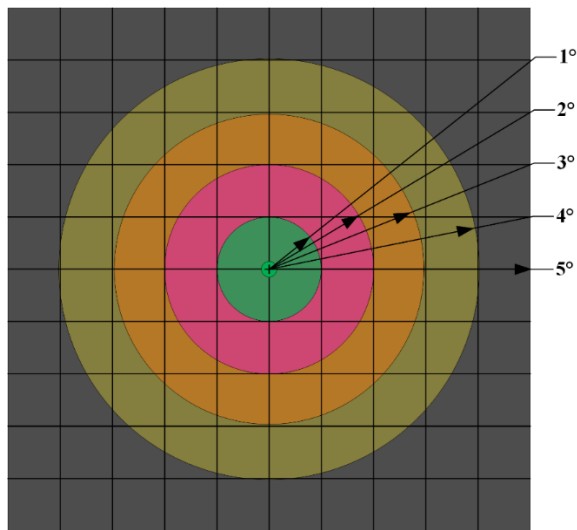


456       Figure A2 shows a schematic diagram illustrating the partitioning of distance between the data along

the track and the reference point. (The green area indicates that the distance between the orbit and the
reference point is less than or equal to 1° and is calculated uniformly as 1°, the pink area indicates
uniformly as 2°, the orange area indicates uniformly as 3°, the brown area indicates uniformly as 4°, and
the black area indicates uniformly as 5°)

## Appendix B: Mutual power spectral density analyses to estimate ocean dynamics variability

464       The coherence function and power spectrum reflect the degree of correlation between the signal and

the density of the power distribution with frequency. The purpose of power spectrum estimation is to
characterize the distribution of the frequency components of signals and stochastic processes based on a
finite data sequence. The coherence function determines the similarity between two repetitive period
signals and thus deduces the resolution capability.
We define the self-coherence function of the random signal $x(t)$ is $R_x(\tau)$ the Fourier transform of
$R_x(\tau)$ is defined as the self-power spectral density of $x(t)$ as shown in Equation (5). The self-power
spectral $S_x(f)$ contains all the information of $R_x(\tau)$.
$$S_x(f) = \int_{-\infty}^{+\infty} R_x(\tau)e^{-j2\pi f\tau}d\tau \qquad (5)$$

where $j$ is the imaginary unit, $f$ is the frequency, $\tau$ is the time delay. The mutual correlation
characteristics of two random signals $x(t)$ and $y(t)$ can be described in the frequency domain by the cross
power spectral density as shown in equation (6). The phase difference obtained by calculating the cross
power spectral density can visualize the degree of similarity between the two signal sequences in the
frequency domain(ZHOU et al.,2024).
$$S_{xy}(f) = \int_{-\infty}^{+\infty} R_{xy}(\tau)e^{-j2\pi f\tau}d\tau \qquad (6)$$

The coherence function of the signals $x(t)$ and $y(t)$ for two repetition cycles is:
$$C_{xy}(f) = \frac{|S_{xy}(f)|^2}{S_x(f)S_y(f)} \qquad (7)$$

Where $S_{xy}(f)$ is the cross spectral density of $x(t)$ and $y(t)$, and $S_x(f)$ and $S_y(f)$ are the self-power spectral densities of $x(t)$ and $y(t)$ respectively. Marks et al. (2016) proposed that the coherence function can be used to assess the similarity between repetitive cycle signals and thus infer the resolving power. They defined the discriminability criterion for geodesic wavelengths as the spatial wavelength corresponding to a mean square coherence of 0.5. In addition, Chelton et al. (2007) conducted a similar study. They defined the scale at which the ratio of the energy spectral density of the AVISO fusion product to the energy spectral density of the along-track data is 1/2 as the minimum resolved wavelength at which the fusion product can recognize eddies. In signal processing, a mean-square agreement of 0.5 usually indicates that half of the power of two signals is correlated in the frequency domain. This not only implies that there is some correlation between the signals, but also indicates that the signals are at this point at a comparable level to the noise. Therefore, when the coherence reaches 0.5, the corresponding wavelength can be regarded as an important threshold for the signal resolution capability. In this study, we set the consistency threshold to 0.5, referring to standard practice in related fields. We define that when the cross-power spectral density correlation of two neighboring cycle trajectories drops to 0.5, it indicates that an ocean dynamic scaling change of at least the corresponding wavelength has occurred in the region. Finally, the correlation between two repetitive cycles of absolute dynamical topography (ADT) is used to determine the scale changes occurring in the ocean, i.e., to analyze the wavelengths of near-time-varying scale changes in the ocean.

Similar to Appendix A, we also divided the globe (0°E ~ 360°E,60°N ~ 60°S) into small 2° × 2° regions, each region was analyzed with an outwardly expanding 10° × 10° box for the mutual power spectrum. Since the along-track data of the two repetition cycles required strict alignment, linear interpolation was required to fill in the missing data. If missing data occurs, to prevent the interpolation from affecting the original ADT signal, we retained only the repetitive tracks with fewer than 15 missing data points. Otherwise, the entire sample was rejected. To enhance the spatial coverage and maximize the number of samples in each box, the along-track data were split into samples of approximately 560 km (280 data points). From the mutual power spectrum, a new parameter was defined: the wavelength at which the mean-square consistency reaches 0.5, indicating the extent of ocean dynamics scale variability at that wavelength. Finally, all wavelengths within each box are weighted and averaged following the same method as in Appendix A to yield the final global between 60°N~60°S distribution. We utilize this parameter to evaluate greater SWOT's ocean resolution capability in Section 4.

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
