# Peer review of "Enhanced resolution capability of SWOT sea surface height measurements and its application in monitoring ocean dynamics variability"

_EGUsphere, 2024_

## Referee Comment (RC3)

The paper applies wavenumber spectral analysis to compare SSH data from four altimetry missions—HY2B, Saral/AltiKa, Sentinel-3A, and SWOT. It evaluates their spectral characteristics and introduces a new method for global statistical analysis. A key focus is on SWOT's ability to resolve small-scale ocean variability, leveraging its finer spatial resolution. The study highlights the differences in spectral content among these missions and attempts to quantify global-scale variations using SWOT data by introducing a new parameter.

While the study presents a well-motivated analysis, the manuscript would benefit from clearer phrasing in several sections to improve readability and eliminate ambiguities. Additionally, certain methodological aspects require further justification, particularly regarding data preprocessing and resolution effects. I recommend that this manuscript be considered for publication, provided the authors address the following major concerns:

**Major Comments**

- The averaging method used to compute spectral slopes (Appendix A), which relies on a distance-weighted scheme, raises concerns about its statistical validity. A detailed justification of this approach, including examples and a discussion of the weight distributions, is necessary to ensure the robustness of the analysis. Additionnaly, A direct comparison with conventional approaches (e.g., standard averaging methods) is needed to show the advantages of its proposed methodology over standard spectral analysis techniques.

- The section "Global Analyses of Ocean Scale Changes" lacks clarity, primarily due to insufficient explanation of key methodological steps. The authors refer to Appendix B, which introduces spectral coherence methods without clearly linking them to the main analysis. The section does not explicitly explain how the proposed global-scale parameter (derived from mutual power spectra) captures ocean variability. The transition from Appendix B's coherence-based method to a global variability assessment is particularly vague.

**Specific Comments**

**Introduction**

**l.28:** "…sub-mesoscale activity can also reverse the cascade of energy from…"

**l. 55**: Xu et al. (2012) instead of Xu Y et al. (2012)

**l. 71**: "*Another altimeter*"
Which one?

**l. 84**: "*reciprocal power spectral analysis*"

The term "reciprocal power spectral density" isn't a standard term, better referring to cross spectral density.

**Datasets and Methodology**

**l. 90-91**: The explanation of how SWOT data is split into along-track components lacks details on the method used. Does this involve simple subsampling, or is interpolation applied?

**l. 92**: *From October to November 2023*
Does it mean October AND November?

**l. 93-94**: *"To compare [...] by SWOT's KaRIn"*
The sentence structure could be improved to convey the intended meaning more effectively.

**l. 103**: "orbit data" rather than "orbit fix data".

**l. 104-108**: There is some confusion in this section due to the repetition of sentences. Is the GDR corrected or uncorrected? What is the difference between a corrected and an uncorrected product? Additionally, is the SGDR corrected or uncorrected?

**l. 116-119**: The advantages of Ka-band in SARAL/AltiKa (lines 116–119) should be supported by relevant literature.

**l. 118**: There seems to be a repetition of the word "water"
A final period is missing between "data" and "The final...".
Altair band: This seems to be a typo. It likely refers to the AltiKa Ka-band, which, if I'm not mistaken, operates exclusively in the Ka-band. Therefore, the phrase 'The final choice was...' may be unnecessary.

**l. 119**: Why were only cycles 175 and 176 selected for SARAL/AltiKa? Do these correspond to the October-November 2023 period?

**l. 120**: *"The Sentinel-3A (S3A) satellite carries the SRAL altimeter…"*
I would specify "…carries Synthetic Aperture Radar Altimeter (SRAL)…"

**l. 127**: As mentioned earlier, please specify why cycles 104 and 105 were selected.

**l. 131**: What are sub-stellar points?

**l. 132**: *"SWOT carries s Ka-band radar"*
Typo: 's' should be changed to 'a'.

**l. 137**: *"Selected data from SWOT's ocean [...] are selected for this paper."*

Remove the first "selected"

**l. 140, 142, 144**: replace "along-orbit" with "along-track"

**l. 140-142**: What kind of corrections are applied?? Clarifying this step is essential for reproducibility.

**l. 149**: *"…Fourier transforming it…"*
I would suggest to write instead: "performing a Fourier transform on it"

**l.154**: *"We calculated the SSH anomalous wavenumber power spectral density (PSD) for each mission…"*
This is unclear. Are the authors referring to the power spectrum of SSH anomalies?

**l. 157**: The phrase "departed from the previous method of averaging" is vague. What method did the authors adopt instead? The preprocessing steps for calculating the PSD in each 10° × 10° box follow methods similar to Dufau et al. (2016).
However, instead of averaging all individual PSDs within a box (as done in previous methods), the authors suggest a different way to compute the average PSD for each box.
This section requires further clarification

**l. 166**: *"… is the 2 km sampling rate…"*
"resolution" instead of "rate".

**l. 169-170**: *"For wavelengths below 25 km for the first three missions, the 1 Hz SSH error level was estimated by fitting a level to the spectrally flat noise levels present in the PSD maps (Figure. 1)."*
This sentence is confusing. I assume Figure 1 presents the unbiased spectra (i.e., without the constant noise level), is that correct? If so, the sentence needs to be rephrased for clarity. I would also suggest adding the noise level to the plot, if possible, to visualize the intersection with the slope. Additionally, including references to Xu and Fu (2012) and Dufau et al. (2016) would be helpful.

**l. 175**: *"…by removing the estimated constant error level below …"*
Remove "below".

**l. 176-178**: *"Diverse methods for calculating the Power Spectral Density (PSD) […] in the estimated PSD slope range"*
I agree but should not be here.

**l. 179**: *"Hence, for the first three conventional missions, we chose wavelengths in the range of 70-250 km"*.
Missing references: Dufau et al., 2016; Le Traon et al., 2008; Xu & Fu, 2011

l. 184-185: "Due to the presence of […] etc (Boas et al., 2022)"
I believe this should not be a sentence on its own, but rather linked to the previous one for better coherence.

l. 185: "wavelength range" instead of "wavelength"

l. 186-188: The one-dimensional mesoscale resolution capability essentially represents the shortest wavelength detectable in along-track altimeter observations where the signal exceeds the noise, correct? Perhaps this sentence could be rephrased for clarity.

**Global resolution capability of altimetry satellites**

l. 207: "[…] theoretical predictions from SQG and GG theories."
Add a reference please.

l. 208-209: The authors should specify the figure number referenced in the cited paper.

l. 212: "[…] oblique pressure tide […]"
Do the authors mean baroclinic instead of oblique?

l. 261: "satellite altimetry observes" instead of "satellites altimetry observed"

l. 226: "for one month" instead of "for the one month".

l. 227: "[…] results of previous studies."
Add references please.

l. 230: "northwest Pacific" instead of "Pacific northwest".

l. 239-241: "However, for the data from the period […] potential error in one of the satellite's corrections"
Could the authors clarify this statement?

l. 243-244: How the authors explain the high-level noise pattern in the mid-north Atlantic for SWOT (Figure 3c)?

l. 256: Dufau et al. (216) instead of Dufau C et al. (2016)".

l. 269: "[…] where the peak slope occurs."
Would it be possible to include a plot to visually represent this statement?

l. 272: "Experiments have demonstrated…"
Add references

**Global analyses of ocean-scale changes**

**l. 284**: As I mentioned earlier, it would be better to refer to the cross-spectral density.

**l. 288**: Marks et al. (2016) instead of Marks K M et al (2016).

**l. 290**: Why 0.5? Add some details please.

**l. 294**: Western Boundary Currents

**l. 297**: northwest Indian Ocean current

**l. 306**: *"world's ocean current regions"*
Remove "world's"

**Appendix B**

**l. 403-407**: How are equations 5 and 6 linked? Is R the cross-correlation function and S the cross-spectral density (Fourier transform of the cross-correlation)? Also, in equation 6, is P equivalent to S? Please be careful with the notation. Additionally, could you define all the variables?

**l. 408-409**: *"The coherence function can judge the [...] cycle signals to infer resolution capability."*
Already mentioned earlier, see l. 397-398.
Marks et al. (2016) instead of Marks K M et al (2016).

**l. 417**: Which kind of interpolation is performed? Linear?

**Figures**

**Figure 1**: The authors should specify whether these spectra are biased or unbiased.

**Figures 2, 3 and 5c**: The units are missing.

**Figure A1**: The legend mentions some orange points, but I cannot see them in the plot.

**References**

**l. 481 and 499**: The reference formatting is inconsistent.

---

## Author Comment (AC1)

**Reply to Comments**

**Table of Contents**

Reviewer   1

**Reviewer**

The article employs wavenumber spectral analysis to conduct experiments and introduces a novel method for global data statistics. It compares data derived from four distinct satellite types, thereby highlighting the advantages of the SWOT satellites. Additionally, the authors analyze global ocean-scale changes by defining a new parameter and utilizing SWOT satellite data. While the proposed methodology and the new parameter are intriguing, they require further clarification and validation. The English writing should be further polished. I recommend that this manuscript be considered for publication, contingent upon addressing the following modifications.

Dear reviewer:

The author's team would like to thank you for reviewing the paper and providing useful feedback and suggestions. We have carefully read and responded to your comments. Your comments are in black font, our explanatory response is in blue font, and the corresponding revision in the manuscript is in red font.

**1. Comment 1**

The paper does not sufficiently demonstrate the significant advantages of the authors' improvements to the algorithm compared to existing methods. To better emphasize the necessity and effectiveness of the proposed enhancements, it is recommended that a comparison with the traditional averaging method be included.

We have noted the shortcomings you pointed out in demonstrating the advantages of the algorithm improvement compared with the existing methods. To better emphasise the necessity and effectiveness of the proposed improvement, we will add a comparison experiment with the traditional averaging method in the revised draft to analyse in detail the difference in detection performance between the two and further verify the advantages of the improved process.

The manuscript was revised as follows.

To compare the difference between the new method of calculating the slope and the previous method, we used the HY2B satellite and calculated the global SSH spectrograms for both methods (Figure 2). We can observe from Figures 2a,2c. When counting the global distribution maps, distance-weighted averaging can reflect the spatial correlation of geographic phenomena more accurately by assigning distance-based weights to the observation points. This method not only reduces the bias caused by local outliers or uneven data distribution, but also enhances the regional representativeness of the distribution map. Figure 2b,2d shows the zoomed-in local area, and it can be clearly observed that the distance-weighted method can more accurately bring out the detailed part of the SSH slope map. In contrast, the traditional equal-weighted averaging method assigns the same weight to all observations, ignoring the effect of distance on the statistical results. This approach tends to lead to excessive smoothing of the signal, especially when analysing subtle changes such as slope, and may mask important local features, thus reducing the accuracy and explanatory power of the distribution plot. Therefore, the use of the distance between trajectories to adjust the weights improves the accuracy and reasonableness of the global distribution statistics and provides a more reliable basis for subsequent analyses.

[Figure]

**Figure. 2 Distribution plots of weighted versus equal weighted averaging (a. Results of weighted averaging using distances between satellite tracks, b. Plot of results averaged using the same weights, c and d are areas enlarged by the black boxes in a and b, respectively)**

**2. Comment 2**

The averaging method presented in Appendix A employs distance-weighted averaging across a range of orbital data. However, it is crucial to assess whether this method is scientifically sound. I request a detailed explanation supported by appropriate examples and a discussion of the statistical weight distribution involved in this approach.

Assessing the scientific validity and reasonableness of the averaging method is critical to the credibility of the study. The averaging method presented in Appendix A is based on calculating the distances between a series of trajectory data and averaging them by weighting the distances. In order to explain the method more clearly, we will describe in detail how the distances were determined and illustrate the principles of weight distribution, while further exploring the distribution of statistical weights.

Further explanation of the method: The core idea of the distance-weighted averaging method is to calculate the distance between the target grid points and the trajectory so as to more accurately reflect the spatial distribution of the data. Specifically, data closer to the target point will be given a higher weight, while data further away from the target point will be given a lower weight. Since the power spectral density of each grid point is calculated based on the average power spectral density of all trajectories within a large $10°\times10°$ area, the grid points are located at a distance of $5°$ from the boundary, as shown in Fig. 1#. We classify the trajectories based on the minimum distance from the trajectory to the grid point.

In addition, considering that we are counting $2°\times2°$ resolution data for the global ocean between $60°N\sim60°S$, when the distance from a trajectory to a grid point is less than or equal to $1°$, we set it to a distance of $1°$ by default, i.e., we assign the same weight to all points within a circle with a radius of less than $1°$ (the green part of Fig. 1#). With each $1°$ increase in circle radius, the corresponding hollow circle region is also assigned the same weight. The black region of the outermost circle has its distance set to $5°$ because it is the furthest away and has a smaller weight share.

$$W_i = \frac{w_i}{\sum_{i=1}^{n} w_i} \qquad\qquad (1)$$

where $w_i$ is $1/D_i$, $W_i$ is the weight of each track involved in the calculation of the PSD. $D_i$ is the distance (°).

[Figure]

图 1#

2. Discussion of statistical weight distribution:

   In order to better demonstrate the scientific validity as well as the necessity of the method, we have counted the distribution of trajectory distances of the four satellites involved in the calculation of the global ice-free regions, respectively. Following the way of defining the distances above, it can be seen from Fig. 2# and Fig. 3#. There are roughly the same number and percentage of satellite trajectories at $10°\times10°$ participating in the calculation with distances from $1°$-$4°$ to the grid points, and a smaller number and percentage with distances of $5°$. This is because the length of the along-track data at the boundary may be smaller than the length of the data used to calculate the power spectral density, resulting in a lower percentage. It can also be demonstrated in Figure. 2# and Figure. 3# that the distance distribution of the trajectories involved in the averaging calculation is large, so distance-weighted averaging is necessary. In practical applications, we believe that the spatial differences between data points of different trajectories can be better handled by such a weighting method, thus improving the accuracy of data analysis.

[Figure]

Figure 2# Distribution of numbers at different distances

Figure 3# Probability distribution at different distances

**3. Comment 3**

Footprints of SWOT, HY2B, Sara and S3A are different. The resolutions of SSHs for these 4 missions are also different. How about the effects of different resolutions on the results?

Different sampling resolutions have a significant effect on the calculated power spectral density. Specifically, the following are the potential effects of different resolutions on the results:

Satellites with higher spatial resolution (e.g. SWOT) are able to capture more details, especially at smaller spatial scales, which results in the power spectral densities of the higher frequency part remaining in line with the predicted trend of Boas et al (2022) (Figure 4#). Satellites with lower spatial resolution (e.g., HY2B), on the other hand, miss some of the smaller-scale variations, and thus the portion of their power spectral densities that follows the same trend as in Fig. 4# is mainly concentrated in the lower-frequency region. The high-frequency part is mainly signals generated by noise, so the PSD is flatter. To summarise, high-resolution satellites provide finer spatial data and are able to capture more details, whereas low-resolution satellites may only capture larger-scale variations, resulting in significant differences between the two mainly in the high-frequency part. The calculated PSD slope plots for satellites with different resolutions are roughly the same, except that the frequency intervals in which the slopes are located are in different ranges. The noise level of each satellite is also different, so the final results of the calculated 1D mesoscale resolution capability are also different.

Reference:

Boas A B V, Lenain L, Cornuelle B D, et al., 2022a. A Broadband View of the Sea Surface Height Wavenumber Spectrum[J]. GEOPHYSICAL RESEARCH LETTERS, 49(4): e2021GL096699.

[Figure]

Figure 4# Schematic SSH wavenumber spectra. (from Villas Boas et al.(2022), Figure 1)

**4. Comment 4**

line 55: Xu Y et al. -> Xu et al.

Thanks to the reviewer for pointing this out. We have amended 'Xu Y et al.' in line 55 to 'Xu et al.' to comply with formatting requirements.'

**5. Comment 5**

line 68: Vergara O et al. -> Vergara et al.

We have amended 'Vergara O et al.' in line 68 to 'Vergara et al.' to comply with formatting requirements.

**6. Comment 6**

section 2.1: Time spans for different satellite altimetry missions are different.

Thank you for your feedback. To avoid misunderstanding, we would like to clarify the period in the paper. The period of the altimetry missions is the same for the different satellites, which may have been misunderstood due to a lack of clarity in our previous explanation. In the experiment, data from all satellites come from 1 October 2023 to 1 November 2023, so the data from the different satellites we use are all partial cycles within that period. We have revised the manuscript accordingly to ensure a more accurate representation of the paper.

The manuscript was revised as follows.

Line109: In this paper, we use SGDR data for HY2B, with a time horizon of October 2023.

Line119: Finally, SARAL/AltiKa data with a time span of October 2023 was selected.

Line127: The data for S3A in October 2023 was selected.

**7. Comment 7**

line 103: What is an uncorrected data product?

Content in the paper is derived from the HY2B datasheet at https://osdds.nsoas.org.cn/home. HY-2B satellite radar altimeter secondary products include Operational Geophysical data record (OGDR),

Interim geophysical data records (IGDR), Sensor geophysical data records (SGDR), and geophysical data records (GDR). Sensor geophysical data records (SGDR), and geophysical data records (GDR). IGDRs are uncorrected data products obtained using MOE orbiting data and waveform reconstruction. The data mainly include effective wave heights, surface wind speeds, sea surface heights, and the relevant correction parameters for the calculation of sea surface heights. IGDR data products are produced and distributed within 55 hours after the reception of the satellite data. GDR is a fully corrected data product obtained by using POE orbiting data and waveform reconstruction methods. The data mainly include effective wave height, sea surface wind speed, sea surface height, and related correction parameters for calculating sea surface height, and the GDR data products are produced within 30 days after satellite data acquisition.

IGDR takes the raw observations and does some preliminary processing, but does not yet perform all the fine corrections, such as precision orbit corrections, atmospheric delay corrections, waveform fitting corrections, etc. These corrections are essential for ensuring the accuracy of the data. These corrections are essential to ensure the accuracy and reliability of the data, so IGDR usually needs to be further processed into a fully corrected data product, such as GDR, before it can be used for scientific research and applications. So IGDR is called uncorrected data.

**8. Comment 8**

line 104: What is an uncorrected data product?

I am very sorry that there was an error in the presentation of this sentence and I was not able to remove the sentence in time due to my carelessness. See comment 7 for a specific explanation.

**9. Comment 9**

line 105: What is an uncorrected data product?

I am very sorry that there was an error in the presentation of this sentence and I was not able to remove the sentence in time due to my carelessness. See comment 7 for a specific explanation.

**10. Comment 10**

line 106: What is a full corrected data product?

See comment 7 for a specific explanation.

**11. Comment 11**

line 113: What about the frequency of SSHs used? 40Hz or 20Hz?

Apologies for the unclear interpretation of the original article. Redundant statements have been deleted. The description is only intended to illustrate the characteristics of the SARAL/AltiKa satellite itself, while the sampling rate of 1hz has been chosen for the conventional one-dimensional satellites in this paper. The sampling rates of the selected satellites are described in the last paragraph of section 2.1.

The along-orbit SSH (HY2B, SARAL/AltiKa, S3A) observations were kept at their original 1hz observation positions at intervals of about 7km and corrected for all instrumental, environmental, and geophysical corrections.

The manuscript was revised as follows.

Using Ka-band allows for reduced size, lower ionospheric attenuation delays, and higher measurement accuracy than conventional Ku-band altimeters (Quartly et al., 2015).

**12. Comment 12**

line 128: The sentence may repeat.

The author has removed redundancies and re-combined sentences to improve clarity of expression.

**13. Comment 13**

line 131: What is the nadir-stellar point?

The paper refers to a 20-km gap centered on the satellite ground track. Words that were not clearly expressed have been reworked.

The manuscript was revised as follows.

SWOT adopts Ka-band radar interferometry (KaRIn) for measurements over a narrow strip of 120 kilometers (a 20-kilometer gap along the track and the centre of this area is sampled by conventional altimeters at a low resolution).

**14. Comment 14**

line 140: along-orbit SSH -> along-track SSH.

We have changed 'long-orbit SSH' to 'long-track SSH' on line 140.

**15. Comment 15**

line 154: What is the SSH anomalous wavenumber power spectral density?

The original article was incorrectly worded and it should have been an SSH anomaly. All power spectral densities calculated later in this paper were calculated by SSH anomaly, also known as SSHA. The manuscript was revised as follows.

We calculated the SSH anomaly wavenumber power spectral density (PSD) for each mission globally in a $10° \times 10°$ box.

**16. Comment 16**

Line 165: What are these three parameters in detail?

Parameter 1 (SSH error level):

As stated by Dufau et al. (2016), the 1 Hz SSH error level is estimated from a horizontal fit to the spectral flat noise level present in the PSD plot for wavelengths below 23.5 km (Figure 5#). The energy level at those wavelengths in the 1Hz SSH PSD corresponds to the sum of the instrumental white noise and a ''hump-shaped spectral artifact''. This artifact is more intense in certain regions because it originates in inhomogeneities in the radar backscatter coefficient within the altimeter footprint leading to an erroneous estimation of the SSH and creating a larger spectral noise. Therefore, the first parameter value of is the error level of the satellite.

Parameter 2(the SSH spectral slope in the mesoscale bands):

Dufau et al. (2016) calculated the PSD slopes by a least squares regression to the spectra for a fixed wavelength band between 95 and 280 km. Diverse methods for calculating the Power Spectral Density (PSD) can result in minor discrepancies in the slope range (Vergara et al., 2019) Additionally, data sampled at varying frequencies may also engender subtle variances in the estimated PSD slope range. Hence, for the first three conventional missions, we chose wavelengths in the range of 70-250 km and fitted the slope of the PSD by least squares. For the SWOT mission, its cross-correction process filters out some noise, resulting in a spectral profile that continually drops, as shown in Figure 6#. Due to the

presence of many sub-mesoscale phenomena at 15-40km, such as internal waves and tides, etc.(Boas et al., 2022). Therefore, a wavelength of 40-125 km was selected for calculating the PSD slope for the SWOT mission.

Parameter 3(one-dimensional mesoscale resolution capability):

The crossing point where the error level and spectral slope intersect, sets the wavelength at which the PSD of the smallest-scale signal is equal to the error level. we call the one-dimensional mesoscale resolution capability.

[Figure]

Figure 5# Mean SSH PSD for 7 months of Jason-2 data in a $10°\times 10°$ box located in the Gulf Stream Current system centered in [294 °E, 39 °N]. The red curve is the unbiased spectrum with the constant noise level removed (horizontal dashed line) from the original spectrum (black curve). The vertical lines are 90% confidence intervals. (from Dufau et al. (2016), Figure A1)

[Figure]

Figure 6# Calculate the spectral range of the slope

Reference:

Dufau C, Orsztynowicz M, Dibarboure G, et al., 2016. Mesoscale resolution capability of altimetry: Present and future[J]. JOURNAL OF GEOPHYSICAL RESEARCH-OCEANS, 121(7): 4910-4927.

Vergara O, Morrow R, Pujol I, et al., 2019. Revised Global Wave Number Spectra From Recent Altimeter Observations[J]. JOURNAL OF GEOPHYSICAL RESEARCH-OCEANS, 124(6):

3523-3537.

Boas A B V, Lenain L, Cornuelle B D, et al., 2022a. A Broadband View of the Sea Surface Height Wavenumber Spectrum[J]. GEOPHYSICAL RESEARCH LETTERS, 49(4): e2021GL096699.

**17. Comment 17**

fig. 2: How to define and determine the slope? This is not global.

Q1: How to define and determine the slope?

In this paper, slope specifically refers to the slope of the power spectral density (PSD) of the along-track data in the mesoscale range. This is shown in Figure 4#. In the mesoscale to sub-mesoscale range of wavenumber, the slope of the power spectral density curve usually behaves as a constant. In order to accurately determine this slope, a least-squares fit based on the relationship between wavenumber and power spectral density was used to obtain a straight line that best fits the data. The coefficient of the primary term of this straight line is the slope sought. It is important to note that the two variables involved in the slope calculation represent the logarithmic values of the wavenumber and the power spectral density (i.e., logarithms with a base of 10), not the actual wavenumber values.

Q2: This is not global.

To avoid further confusion, we have reworked the presentation and made it clear that it is region-specific data, not global results.

The manuscript was revised as follows.

Accordingly, this paper presents an updated analysis of the global SSH spectral slope between 60 °N and 60 °S, using data from various altimetry missions.

**18. Comment 18**

fig. 3: This is not global.

To avoid further confusion, we have reworked the presentation and made it clear that it is region-specific data, not global results.

The manuscript was revised as follows.

**Figure. 4 Noise levels of different satellites ((a) HY2B, (b) SARAL/ALTIKA, (c) S3A, (d) SWOT)**

**19. Comment 19**

fig. 4: This is not global.

To avoid further confusion, we have reworked the presentation and made it clear that it is region-specific data, not global results.

The manuscript was revised as follows.

**Figure. 5 Distribution of eddy kinetic energy calculated by MDT2022**

**20. Comment 20**

fig. 5: This is not global. In line 275, the colorbar units in Figure 5c are absent. Please ensure that the units are indicated for clarity and proper interpretation of the data.

We have made the following changes to the issues you raised:

fig. 5: This is not global. The beginning of the paper is already redefining 'global'. We have amended the relevant statement in the manuscript to ensure greater clarity on this point.

Units for Figure 5c: We have added missing units to the colorbar in Figure 5c to ensure clarity and correct

interpretation of the data.

The manuscript was revised as follows.

[Figure]

**Figure. 6 Resolved wavelengths for different satellites (a. HY2B, b. SARAL/ALTIKA, c. S3A, d. SWOT)**

**21. Comment 21**

Section 4 introduces a new parameter; however, the experiments are conducted exclusively using data from the SWOT 21-day repeat cycles. Is it feasible to conduct experiments using data from the SWOT one-day repetition cycles for specific locations? This approach would more effectively illustrate SWOT's contributions to understanding ocean sub-mesoscale dynamics.

We are grateful to the reviewers for their in-depth consideration of our research methodology. Regarding the feasibility of using site-specific SWOT one-day repeat cycle data for experiments, we believe this is a very valuable suggestion. In the current study, we used SWOT 21-day repetitive cycle data mainly because it can cover a wider spatial range, which helps to fully assess ocean dynamics scale variability. However, experiments using SWOT one-day repeat cycle data do help to provide insights into analysing ocean dynamics scale variability over short periods of time, in particular the contribution to sub-mesoscale dynamics. We will add to the original manuscript a discussion of the use of SWOT one-day repetition cycles. and explore how it can further improve the precision of our study and our understanding of sub-mesoscale dynamics.

The manuscript was revised as follows.

To better illustrate the role of SWOT one-day repetitive cycle data, we utilised two repetitive cycle trajectory data, SWOT_L3_LR_SSH_Expert_478_021_20230402T142629_20230402T151735_v1.0.nc and SWOT_L3_LR_SSH_ Expert_479_021_20230403T141706_20230403T150812_v1.0.nc. During the analysis, we divided the SWOT data into two strips along the track direction for the calculation. Of the multiple along-track data within each strip, we selected 100 data points as the data length for calculating the reciprocal power spectrum. The calculation results for each strip were obtained by averaging the data from multiple along-track directions. As shown in Figure 8, the reciprocal power spectra computed using the same pass data on these two days reveal the scale differences in the ocean dynamics variations at different locations. Figures 8b, 8c and 8d, 8e show the magnified images of sea surface height anomalies at the same locations on 2 April 2023 and 3 April 2023 for regions 1 and 2 in Figure 8a, respectively. As can be seen from Figure 8a, region 2 experienced a scale change of about 30-35 km wavelength in just two days, while region 1 had a smaller scale change of about 20-25 km. This suggests that there are significant differences in the scale of ocean dynamics in different regions, even within the same time period. In addition, within the same region, the variability varies between strips,

mainly because different regions are affected by different sub-mesoscale phenomena. Sub-mesoscale dynamics variability in the ocean is constantly occurring, which leads to differences between the left and right SWOT strips. However, this experiment only considered the direction along the satellite track and analysed the scale variation of ocean dynamics in the one-dimensional direction along the track. At the same time, the averaging of data from multiple along-track directions may diminish the significance of ocean dynamics scale variations within a local region. The aim is to provide a feasible framework for subsequent related studies and to lay the foundation for more in-depth ocean dynamics studies. Subsequent work will continue to be devoted to the study of ocean dynamics scale variability in two dimensions using SWOT.

[Figure]

**Figure. 8 Ocean dynamics scale changes calculated using SWOT one-day repeat cycle data**

**22. Comment 22**

How to show the dynamic mechanism?

By 'dynamic mechanism' we mean the scale of ocean dynamics that occurs in a given region over a given time frame. By analysing the reciprocal power spectra of two data cycles, we aim to assess the correlation between the data and infer the ocean dynamics characteristics of the region. A high correlation between the data of the two cycles indicates that the scale of the ocean-dynamic phenomena is small, while a poor correlation may imply that ocean-dynamic phenomena are occurring at a larger scale in the region. By setting a correlation threshold of 0.5, we were able to determine that the region would have at least that wavelength scale of dynamical changes. Based on this idea, we propose the parameter 'ocean dynamics scale variability', which is used to describe the minimum scale of ocean dynamics variability that occurs within a certain period worldwide between 60°N~60°S. We hope that this interpretation can clarify the meaning of the term 'ocean dynamics scale variability'.

It is hoped that this explanation will clarify the definition and application of 'dynamic mechanism' in this study.

**23. Comment 23**

fig.6: This is not global.

To avoid further confusion, we have reworked the presentation and made it clear that it is region-specific data, not global results.

The manuscript was revised as follows.
**Figure. 7 Ocean mesoscale and sub-mesoscale scale changes over four seasons**

**24. Comment 24**

The reference cited at the end of line 72 is incorrect, and the reference formatting at the end of line 55 is inconsistent. Please review the reference formatting throughout the manuscript to ensure uniformity.

Thank you to the reviewers for their careful review of the reference format. We have made the following changes based on your suggestions:

References in line 72: we have verified and corrected the references cited at the end of line 72 to ensure that they are accurate.

Line 55 reference formatting: we have standardised the formatting of the reference at the end of line 55 to make it consistent with the other references.

We have thoroughly checked and standardised the formatting of all references in the manuscript to ensure consistent formatting.

---

## Author Comment (AC2)

**Reply to Comments**

**Table of Contents**

**Reviewer2**

The paper applies wavenumber spectral analysis to compare SSH data from four altimetry missions—HY2B, Saral/AltiKa, Sentinel-3A, and SWOT. It evaluates their spectral characteristics and introduces a new method for global statistical analysis. A key focus is on SWOT's ability to resolve small-scale ocean variability, leveraging its finer spatial resolution. The study highlights the differences in spectral content among these missions and attempts to quantify global-scale variations using SWOT data by introducing a new parameter.

While the study presents a well-motivated analysis, the manuscript would benefit from clearer phrasing in several sections to improve readability and eliminate ambiguities. Additionally, certain methodological aspects require further justification, particularly regarding data preprocessing and resolution effects. I recommend that this manuscript be considered for publication, provided the authors address the following major concerns:

Dear reviewer:

The author's team would like to thank you for reviewing the paper and providing useful feedback and suggestions. We have carefully read and responded to your comments. Your comments are in black font, our explanatory response is in blue font, and the corresponding revision in the manuscript is in red font. In response to your review comments, we name the Figure with #.

**Major Comments**

**Comment1**

The averaging method used to compute spectral slopes (Appendix A), which relies on a distance-weighted scheme, raises concerns about its statistical validity. A detailed justification of this approach, including examples and a discussion of the weight distributions, is necessary to ensure the robustness of the analysis. Additionally, a direct comparison with conventional approaches (e.g., standard averaging methods) is needed to show the advantages of its proposed methodology over standard spectral analysis techniques.

**Question 1:A detailed justification of this approach, including examples and a discussion of the weight distributions, is necessary to ensure the robustness of the analysis.**

The core idea of the distance-weighted averaging method is to calculate the distance between the target grid points and the trajectory to more accurately reflect the spatial distribution of the data. Specifically, data closer to the target point will be given a higher weight, while data further away from the target point will be given a lower weight. Since the power spectral density of each grid point is calculated based on the average power spectral density of all trajectories within a large $10° × 10°$ area, the grid points are located at a distance of 5° from the boundary, as shown in Figure 1#. We classify the trajectories based on the minimum distance from the trajectory to the grid point. In addition, when the distance from a trajectory to a grid point is less than or equal to 1°, we set it to a distance of 1° by default, i.e., we assign the same weight to all points within a circle with a radius of less than 1° (the green part of Figure 1#). With each 1° increase in circle radius, the corresponding hollow circle region is also assigned the same weight. The black region of the outermost circle has its distance set to 5° because it is the furthest away and has a smaller weight share.

$$W_i = \frac{w_i}{\sum_{i=1}^n w_i} \qquad (1)$$

Where $w_i$ is $1/D_i$, $W_i$ is the weight of each track involved in the calculation of the PSD. $D_i$ is the distance (°).

[Figure]

Figure 1# Criteria for the definition of distances between satellite tracks

To validate the methodological rigor and operational efficacy of our approach, a systematic analysis was conducted on the spatial distribution characteristics of trajectory distances from four satellites involved in calculating ice-free regions globally. As illustrated in Figures 2 and 3, which adhere to the predefined distance classification criteria, satellite trajectories within 10°×10° grid boxes exhibited roughly the same number and percentage of distributions across 1°-4° distance intervals, while demonstrating significantly reduced percentage at the 5° threshold. This is because the length of the data along the trajectory for the portion at the boundary is less than the length of the data used to calculate the power spectral density, resulting in less data involved in the averaging and thus a lower percentage. It can also be demonstrated in Figure 2# and Figure 3# that the distance distribution of the trajectories involved in the averaging calculation is large, so distance-weighted averaging is necessary. In practical applications, we believe that the spatial differences between data points of different trajectories can be better handled by such a weighting method, thus improving the accuracy of data analysis.

[Figure]

Figure 2# Amount of data matching grid points to satellite tracks in all 10° x 10° boxes.

[Figure]

Figure 3# Ratio of data matching grid points to satellite tracks in all 10° x 10° boxes.

**Question 2 : A direct comparison with conventional approaches (e.g., standard averaging methods) is needed to show the advantages of its proposed methodology over standard spectral analysis techniques.**

We have taken note of the shortcomings you highlighted regarding the demonstration of the advantages of our algorithm improvement compared to existing methods. To more effectively emphasize the necessity and effectiveness of the proposed improvement, we will incorporate an additional comparison experiment with the traditional averaging method in the revised manuscript. This experiment will provide a detailed analysis of the differences in detection performance between the two methods and further validate the superiority of the improved process.

The manuscript was revised as follows.

To compare the difference between the new method of calculating the slope and the previous method, we used the HY2B satellite and calculated the global SSH spectrograms for both methods (Figure 2). We can observe from Figures 2a,2c. When counting the global distribution maps, distance-weighted averaging can reflect the spatial correlation of geographic phenomena more accurately by assigning distance-based weights to the observation points. This method not only reduces the bias caused by local outliers or uneven data distribution but also enhances the regional representativeness of the distribution map. Figure 2b,2d shows the zoomed-in local area, and it can be observed that the distance-weighted method can more accurately bring out the detailed part of the SSH slope map. In contrast, the traditional equal-weighted averaging method assigns the same weight to all observations, ignoring the effect of distance on the statistical results. This approach tends to lead to excessive smoothing of the signal, especially when analysing subtle changes such as slope and may mask important local features, thus reducing the accuracy and explanatory power of the distribution plot. Therefore, the use of the distance between trajectories to adjust the weights improves the accuracy and reasonableness of the global distribution statistics and provides a more reliable basis for subsequent analyses.

[Figure]

**Figure 2 Distribution plots of weighted versus equal-weighted averaging (a. Results of weighted averaging using distances between satellite tracks, b. Plot of results averaged using the same weights, c, and d are areas enlarged by the black boxes in a and b, respectively)**

**Comment2**

The section "Global Analyses of Ocean Scale Changes" lacks clarity, primarily due to insufficient explanation of key methodological steps. The authors refer to Appendix B, which introduces spectral coherence methods without clearly linking them to the main analysis. The section does not explicitly explain how the proposed global-scale parameter (derived from mutual power spectra) captures ocean variability. The transition from Appendix B's coherence-based method to a global variability assessment is particularly vague.

Thank you for your comments. In response to several of your questions, the authors have provided further explanations and additions to the relevant content. The details are as follows::

**1. The Issue relating to the methodological and principal analytical linkages in Appendix B.**

In Appendix B of the original manuscript, the authors omitted an equation and the naming of variables was not further standardized. Therefore, we have re-described the method and standardized the relevant variables, labeling each parameter in the formula. The manuscript is revised as follows.

We define the self-coherence function of the random signal $x(t)$ as $R_x(\tau)$ the Fourier transform of $R_x(\tau)$ is defined as the self-power spectral density of $x(t)$ as shown in Equation (5). The self-cross power spectral density $S_x(f)$ contins all the information of $R_x(\tau)$,.

$$S_x(f) = \int_{-\infty}^{+\infty} R_x(\tau)e^{-j2\pi f\tau}d\tau \tag{5}$$

where $j$ is the imaginary unit, $f$ is the frequency, $\tau$ is the time delay. The mutual correlation characteristics of two random signals $x(t)$ and $y(t)$ can be described in the frequency domain by the cross power spectral density as shown in equation (6). The phase difference obtained by calculating the cross power spectral density can visualize the degree of similarity between the two signal sequences in the frequency domain(ZHOU et al.,2024).

$$S_{xy}(f) = \int_{-\infty}^{+\infty} R_{xy}(\tau)e^{-j2\pi f\tau}d\tau \tag{6}$$

The coherence function of the signals $x(t)$ and $y(t)$ for two repetition cycles is:

$$C_{xy}(\omega) = \frac{|S_{xy}(f)|^2}{S_x(f)S_y(f)} \tag{7}$$

Where $S_x(f)$ is the cross spectral density of $x(t)$ and $y(t)$, and $S_x(f)$ and $S_y(f)$ are the self-power spectral densities of $x(t)$ and $y(t)$ respectively.

**2. On the transition from Appendix B to the global variability assessment.**

The process on how to capture scales of ocean dynamics change from the correlation of cross power spectral densities is as follows. First, Marks et al. (2016) proposed that the coherence function can be used to assess the similarity between repetitive cycle signals and thus infer the resolving power. They defined the discriminability criterion for geodesic wavelengths as the spatial wavelength corresponding to a mean square coherence of 0.5. In addition, Chelton et al. (2007) conducted a similar study. They defined the scale at which the ratio of the energy spectral density of the AVISO fusion product to the energy spectral density of the along-track data is 1/2 as the minimum resolved wavelength at which the fusion product can recognize eddies. In signal processing, a mean-square agreement of 0.5 usually indicates that half of the power of two signals is correlated in the frequency domain. This not only implies that there is some correlation between the signals but also indicates that the signals are at this point at a comparable level to the noise. Therefore, when the coherence reaches 0.5, the corresponding wavelength can be regarded as an important threshold for the signal resolution capability.

In this study, we utilize along-track ADT data from two adjacent cycles to calculate the correlation of their power spectral densities. Ideally, if the two neighboring cycle tracks are identical, the correlation should be 1. However, the ocean is a dynamic environment with complex ocean dynamics scale variability occurring all the time at the sea surface. These changes make the spectra of two neighboring periodic trajectories not identical but show some similarity. Therefore, referring to standard practice in related fields, we define that when the cross-power spectral density correlation of two neighbouring cycle trajectories drops to 0.5, it indicates that ocean dynamics scale variability of at least the corresponding wavelength is occurring in this region. Based on this parameter, we conducted a correlation analysis of the 21-day repeated orbit data and the 1-day repeated orbit data of the SWOT satellite and drew relevant conclusions.

The original manuscript was revised and supplemented as follows. (Original manuscript 409 lines)

Marks et al. (2016) proposed that the coherence function can be used to assess the similarity between repetitive cycle signals and thus infer the resolving power. They defined the discriminability criterion for geodesic wavelengths as the spatial wavelength corresponding to a mean square coherence of 0.5. In addition, Chelton et al. (2007) conducted a similar study. They defined the scale at which the ratio of the energy spectral density of the AVISO fusion product to the energy spectral density of the along-track data is 1/2 as the minimum resolved wavelength at which the fusion product can recognize eddies. In signal processing, a mean-square agreement of 0.5 usually indicates that half of the power of two signals is correlated in the frequency domain. This not only implies that there is some correlation between the signals, but also indicates that the signals are at this point at a comparable level to the noise. Therefore, when the coherence reaches 0.5, the corresponding wavelength can be regarded as an important threshold for the signal resolution capability. In this study, we set the consistency threshold to 0.5, referring to standard practice in related fields. We define that when the cross-power spectral density correlation of two neighboring cycle trajectories drops to 0.5, it indicates that an ocean dynamic scaling change of at least the corresponding wavelength has occurred in the region. Finally, the correlation between two repetitive cycles of absolute dynamical topography (ADT) is used to determine the scale changes occurring in the ocean, i.e., to analyze the wavelengths of near-time-varying scale changes in the ocean.

**Reference**

Chelton, D. B., Schlax, M. G., Samelson, R. M., & de Szoeke, R. A. (2007). Global observations of large oceanic eddies. Geophysical Research Letters, 34(15).

**Specific Comments**

**Comment1**

l.28: "…sub-mesoscale activity can also reverse the cascade of energy from…"

Thanks to the reviewer's careful review, we have revised this sentence.

The manuscript was revised as follows.

Driven by different mechanisms of quasi-geostrophic (QG) dynamics, **sub-mesoscale activity can also reverse the cascade of energy from** the sub-mesoscale to the mesoscale energy (Cao et al., 2021; Qiu et al.,2022).

**Comment2**

l. 55: Xu et al. (2012) instead of Xu Y et al. (2012)

Thanks to the reviewer for pointing this out. We have amended 'Xu Y et al.' in line 55 to 'Xu et al.' to comply with formatting requirements.

**Comment3**

l. 71: "Another altimeter" Which one?

The author did not make this clear in the original article. What the author was trying to convey is that the Sentinel-3A altimeter is using the Ku-band SAR mode.

The manuscript was revised as follows.

**The altimeter of Sentinel-3A** uses the Ku-band synthetic aperture radar (SAR) mode, which achieves lower noise compared to Jason2 and Saral/AltiKa. (Vergara et al., 2019).

**Comment4**

l. 84: "reciprocal power spectral analysis" The term "reciprocal power spectral density" isn't a standard term, better referring to cross spectral density.

Thanks to the reviewers for their valuable comments. In the original paper, the expression "reciprocal power spectral analysis" is not precise enough and can easily cause misunderstanding. Cross spectral density is a standard term used to describe the power spectral correlation between two signals. Therefore, we have corrected the relevant terms in the revised manuscript.

The manuscript was revised as follows (the remaining parts have also been replaced).

Section 4 defines a parameter using cross power spectral analysis and analyzes worldwide between 60°N and 60°S ocean dynamics variability at the mesoscale and sub-mesoscale using SWOT data.

**Comment5**

l. 90-91: The explanation of how SWOT data is split into along-track components lacks details on the method used. Does this involve simple subsampling, or is interpolation applied?

In this paper, we are simply splitting the 2D data from SWOT along the direction of the satellite trajectory into many 1D along-track data and selecting two of them. I think this falls under the category

of subsampling.

Detailed description:

SWOT provides two-dimensional sea surface height observation data. To ensure the physical properties of the data and avoid interpolation errors, we adopt the following processing strategy: firstly, we split the original two-dimensional grid data into 69 independent one-dimensional along-track data (69 points across the track for two-dimensional data), preserving their original signal characteristics intact. In the experimental design stage, to improve the computational efficiency and verify the feasibility of the method, we select the 15th and 45th along-track data as representative samples. The spacing between the two selected along-track data is about 60 kilometers, and the spacing between them and the 15th and 45th of the neighboring orbits of SWOT satellites (shown in Figure 4#) is roughly the same, which can effectively support the research demand of 2°×2° spatial scale;

[Figure]

Figure 4# SWOT data track map after subsampling

The manuscript was revised as follows.

SWOT provides two-dimensional sea surface height observations. We subsample the original two-dimensional gridded data and split it into 69 separate one-dimensional along-track data (That is, the two-dimensional data has 69 cross-track points), and selected the 15th and 45th of these as experimental data.

**Comment6**

l. 92: From October to November 2023: Does it mean October AND November?

The author originally intended "From October to November 2023" to indicate the full month of October, which was indeed an expression error, thanks for pointing it out, and we've corrected it. The manuscript was revised as follows.

For Section 3, we only selected data from October 2023 for analysis due to the large amount of SWOT data.

**Comment7**

l. 93-94: "To compare […] by SWOT's KaRIn": The sentence structure could be improved to convey the intended meaning more effectively.

Thanks for the valuable suggestions, we have made the following changes.

The manuscript was revised as follows.

The resolution capabilities of different altimetry techniques are compared to validate the higher resolution enhancements brought by SWOT's KaRIn.

**Comment8**

l. 103: "orbit data" rather than "orbit fix data".

In line 103 of the text, the author has replaced "orbit fix data" with "orbit data".

**Comment9**

l. 104-108: There is some confusion in this section due to the repetition of sentences. Is the GDR corrected or uncorrected? What is the difference between a corrected and an uncorrected product? Additionally, is the SGDR corrected or uncorrected?

**Question 1: Sentence repetition problems**

The author has removed redundancies and re-combined sentences to improve clarity of expression.

**Question 2: Is the GDR corrected or uncorrected?**

GDR is a fully corrected data product obtained by using POE orbiting data and waveform reconstruction methods. The data mainly include effective wave height, sea surface wind speed, sea surface height, and related correction parameters for calculating sea surface height, and the GDR data products are produced within 30 days after satellite data acquisition.

**Question 3: What is the difference between a corrected and an uncorrected product?**

IGDR takes the raw observations and does some preliminary processing, but does not yet perform all the fine corrections, such as precision orbit corrections, atmospheric delay corrections, waveform fitting corrections, etc. These corrections are essential for ensuring the accuracy of the data. These corrections are essential to ensure the accuracy and reliability of the data, so IGDR usually needs to be further processed into a fully corrected data product, such as GDR, before it can be used for scientific research and applications. So IGDR is called uncorrected data.

**Question 4: is the SGDR corrected or uncorrected?**

SGDR is the same as GDR, but the difference lies in including waveform data. Therefore, SGDR is also fully corrected data.

The manuscript was revised as follows.

The HY2B satellite mission Level 2 products are all released by the National Satellite Ocean Application Service Center of China (NSOAS, http://www.nsoas.gov.cn/), with a repeat cycle of 14 days. The satellite mainly carries dual-frequency radar altimeter (Ku and C bands), and the Ku band is mainly used for distance measurement. HY-2B satellite radar altimeter secondary products include Operational Geophysical Data Records (OGDR), Interim Geophysical Data Records (IGDR), Sensor Geophysical Data Records (SGDR), and Geophysical Data Records (GDR). IGDR is an uncorrected data product obtained using Medium Orbit Ephemeris (MOE) orbit data, waveform reconstruction, etc. GDR is a fully

corrected data product obtained using Precise orbit ephemeride (POE) orbit data, waveform reconstruction, etc. SGDR is the same as GDR, but the difference lies in including waveform data. In this paper, we use SGDR data for HY2B, with a time horizon of October 2023.

**Comment10**

l. 116-119: The advantages of Ka-band in SARAL/AltiKa (lines 116–119) should be supported by relevant literature.

Based on your comments, we conducted further searches of the relevant literature and added the following literature support to the revised manuscript:

The advantages of the Ka-band are reduced ionospheric effects, smaller footprint, better horizontal resolution, and higher vertical resolution **(Verron et al., 2015; Smith et al., 2015)**. A disadvantage of the Ka-band is the attenuation in rainy conditions due to water or vapor and the resultant loss of data **(Lillibridge et al., 2014)**.

**Comment11**

l. 118: There seems to be a repetition of the word "water". A final period is missing between "data" and "The final...". Altair band: This seems to be a typo. It likely refers to the AltiKa Ka-band, which, if I'm not mistaken, operates exclusively in the Ka-band. Therefore, the phrase 'The final choice was...' may be unnecessary.

Thank you very much for your careful review and valuable comments on line 118. We have scrutinized and amended the relevant issues and these amendments will make our text more accurate and more clear. If you have any other comments or suggestions, we stand ready to make further improvements.

The manuscript was revised as follows.

A disadvantage of the Ka-band is the attenuation in rainy conditions due to water or vapor and the resultant loss of data **(Lillibridge et al., 2014).** Finally, SARAL/AltiKa data with a period of October 2023 was selected.

**Comment12**

l. 119: Why were only cycles 175 and 176 selected for SARAL/AltiKa? Do these correspond to the October-November 2023 period?

In the manuscript, we filtered the data according to the criterion of time horizon, which happened to be within these two cycles, and the following changes were made to avoid misrepresentation.

The manuscript was revised as follows.

Finally, SARAL/AltiKa data with a period of October 2023 was selected.

**Comment13**

l. 120: "The Sentinel-3A (S3A) satellite carries the SRAL altimeter…" : I would specify "…carries Synthetic Aperture Radar Altimeter (SRAL)…"

The issue of the full name "SRAL" that you pointed out does help to enhance the clarity and professionalism of the text. In response to your comment, we have clarified in the text that "SRAL" is a "Synthetic Aperture Radar Altimeter".

The manuscript was revised as follows.

The Sentinel-3A (S3A) satellite carries the **Synthetic Aperture Radar Altimeter (SRAL)** for distance

measurements, which is processed using delayed Doppler processing designed to achieve significantly higher signal-to-noise ratios (Heslop et al., 2017).

**Comment14**

l. 127: As mentioned earlier, please specify why cycles 104 and 105 were selected.

To improve the clarity of the paper, we have revised the relevant parts to provide a more precise and coherent explanation.

The manuscript was revised as follows.

The data for S3A in October 2023 was selected.

**Comment15**

l. 131: What are sub-stellar points?

The paper refers to a 20-km gap centered on the satellite ground track. Words that were not clearly expressed have been reworked.

The manuscript was revised as follows.

SWOT adopts Ka-band radar interferometry (KaRIn) for measurements over a wide swath of 120 kilometers (a nadir 20-km gap is supplementally measured by a conventional altimeter at a low resolution).

**Comment16**

l. 132: "SWOT carries s Ka-band radar" : Typo: 's' should be changed to 'a'.

We have corrected this in the revised draft by changing "SWOT carries s Ka-band radar" to "SWOT carries a Ka-band radar".

**Comment17**

l. 137: "Selected data from SWOT's ocean […] are selected for this paper." Remove the first "selected"

Thank you for pointing out the redundancy in the text. We have deleted the first "selected".

**Comment18**

l. 140, 142, 144: replace "along-orbit" with "along-track"

We have changed 'long-orbit SSH' to 'long-track SSH' on lines 140, 142, 144.

**Comment19**

l. 140-142: What kind of corrections are applied?? Clarifying this step is essential for reproducibility.

The corrections of the HY2B satellite are mainly made according to the data editing guidelines given in the satellite user's manual, and they are recognized and corrected during the satellite data pre-processing. The correction items mainly include dry troposphere correction, wet troposphere correction, ionosphere correction, sea state correction, ocean tide correction, solid earth tide correction, polar tide correction, and atmospheric reverse pressure correction. The user manual is available at: https://osdds.nsoas.org.cn/home. Table 1 shows the thresholds for each correction term of HY2B.

Table 1 HY-2B Correction Parameters for Use

| Correction term parameters | model | Limit (unit: m) |
| --- | --- | --- |
| dry troposphere correction | NCEP | -2.4~-2.1 |
| wet troposphere correction | NCEP | -0.5~0.001 |

| | | |
|---|---|---|
| ionosphere correction | GIM | -0.2～0.2 |
| sea state correction | empirical solution | -0.5～0.2 |
| ocean tide correction | GOT4.10c | -3～3 |
| solid earth tide correction | Cartwright and Tayler tables | -0.25～0.1 |
| polar tide correction | Wahr | -0.01～0.01 |
| atmospheric reverse pressure correction | NCEP LEGOS/CN | -0.4～0.8 |
| High-frequency in sea surface topography | LEGOS/CNES | -0.1～0.1 |

The SARAL/AltiKa and S3A satellites utilize a Level 2 sea surface height anomaly product provided by AVISO. (SARAL/AltiKa:https://tds-odatis.aviso.altimetry.fr/thredds/catalog/dataset-l2-geophysical-data-record-saral-ssha-gdr-f/catalog.html;

S3A: Catalog http://tds.aviso.altimetry.fr/thredds/catalog/dataset-l2p-uncross-calibrated-ntc-sla-s3a-1hz/catalog.html) These two products have undergone similar pre-processing steps as the HY2B satellites.

The manuscript was revised as follows.

The along-track SSH (HY2B, SARAL/AltiKa, S3A) observations were kept at their original 1hz observation positions at intervals of about 7km and corrected for environmental, and geophysical corrections. The corrections of the HY2B satellite are mainly made according to the data editing guidelines given in the satellite user's manual, and they are corrected during the satellite data pre-processing. The correction kinds mainly include dry troposphere correction, wet troposphere correction, ionosphere correction, sea state correction, ocean tide correction, solid earth tide correction, polar tide correction, and atmospheric inverse pressure correction. The other two satellites, SARAL/AltiKa and S3A, were directly fed through AVISO's Level 2 sea surface height anomaly product. Both products undergo a similar pre-processing step as the HY2B satellite.

**Comment20**

l. 149: "…Fourier transforming it…" I would suggest to write instead: "performing a Fourier transform on it"

We have revised the original text as you suggested.

The manuscript was revised as follows.

The spectral signal is obtained by sampling the signal in the time domain and **performing a Fourier transform on it**, then sampling it in the frequency domain to obtain a frequency domain signal.

**Comment21**

l.154: "We calculated the SSH anomalous wavenumber power spectral density (PSD) for each mission…" This is unclear. Are the authors referring to the power spectrum of SSH anomalies?

Thank you for your correction on line 154. The description of "SSH anomalous wavenumber power spectral density" mentioned by you is indeed not clear enough, which may easily lead to misunderstanding. According to your suggestion, we have revised and supplemented the relevant description to more accurately express the content of our study. In the original article, "SSH anomalous wavenumber power spectral density" indeed refers to the power spectral density (PSD) obtained by Fourier transforming the along-track sea surface height (SSH) anomalous data of the satellite.

To express this point more clearly, we have modified the original manuscript as follows.

We performed a Fourier transform on the along-track SSH anomaly data for each mission in a 10°x10° box and calculated its wavenumber power spectral density (PSD).

**Comment22**

l. 157: The phrase "departed from the previous method of averaging" is vague. What method did the authors adopt instead? The preprocessing steps for calculating the PSD in each 10°x10° box follow methods similar to Dufau et al. (2016). However, instead of averaging all individual PSDs within a box (as done in previous methods), the authors suggest a different way to compute the average PSD for each box. This section requires further clarification

Thanks to the reviewer for pointing out this issue. We understand the reviewer's confusion about the expression "departed from the previous method of averaging", and hereby provide a detailed explanation of the relevant methods. In this study, we did refer to the method of Dufau et al. (2016) for the PSD data within each 10° × 10° box. However, instead of simply same weighting all individual PSDs within each box to calculate the average, we propose a new approach based on trajectory distance weighting to calculate the average PSD for each box. Specifically, the following steps are included:

**Weight assignment:** unlike the previous method of averaging directly, we assign weights to each individual PSD. The weights are assigned based on the distance of the trajectory from the reference point. For example, we assign higher weights to PSDs whose trajectories are close to the reference point, and lower weights to PSDs whose trajectories are far from the point to be sought.

**Weighted average calculation:** After the weights are assigned, we use the distance weighted average method to calculate the average PSD of each box, which can more accurately reflect the real distribution of PSDs in the boxes, and avoid the influence of individual anomalies on the overall average value.

The manuscript was revised as follows (The bolded red font is the revised manuscript).

**Although we want to obtain results with a resolution of 2° x 2° between 60°N and 60°S globally, each grid point will be expanded into a 10° x 10° box for calculating statistics. We performed a Fourier transform on the along-track SSH anomaly data for each mission in a box and calculated its wavenumber power spectral density (PSD).** The specific preprocessing steps for calculating the PSD for each along-track SSH within a box are similar to those described by Dufau et al.(2016). **For each 10° × 10° box, instead of simply assigning the same weighting to all individual PSDs within each box to calculate the average, we propose a new approach based on trajectory distance weighting to calculate the average PSD for each box. This is because averaging over a 10° × 10° area would diminish the signal in regions with higher mesoscale energies, as well as affect the assessment of areas with lower energies., leading to larger errors in the results.** Consequently, we follow the method in Appendix A to calculate the distance between each SSH along the track and the reference point. Then, the weight of the PSD for each SSH in the region is assigned based on this distance. The method of weighted averaging can reduce the error in calculating the PSD of the grid points and preserve the signal of the grid point location as much as possible so that the calculation results can be more credible.

**Comment23**

l. 166: "... is the 2 km sampling rate…" : "resolution" instead of "rate".

We have changed 'sampling rate' to 'sampling resolution on line 166.

**Comment24**

l. 169-170: "For wavelengths below 25 km for the first three missions, the 1 Hz SSH error level was estimated by fitting a level to the spectrally flat noise levels present in the PSD maps (Figure 1)." This

sentence is confusing. I assume Figure 1 presents the unbiased spectra (i.e., without the constant noise level), is that correct? If so, the sentence needs to be rephrased for clarity. I would also suggest adding the noise level to the plot, if possible, to visualize the intersection with the slope. Additionally, including references to Xu and Fu (2012) and Dufau et al. (2016) would be helpful.

Thanks to the reviewers for their comments. For the spectra of the first three altimetry missions (ALTIKA, HY2B, S3A), we took the average PSD value corresponding to wavelengths below 25 km as a constant level of noise. The spectra of SWOT satellites, on the other hand, have been trending downward and do not show a flatter spectrum. This is because we chose the L3 data product, which is self-cross-corrected by the satellite and cross-corrected by other missions. This means that SWOT's L3 product indirectly removes noise, resulting in a consistent downward trend in the spectrum. We argue that the noise of SWOT mainly exists below 15km based on Chelton et al. (2019). Therefore, for the SWOT task, we take the average value of PSD corresponding to wavelengths below 15 km as the constant level of noise. The biased spectra are shown in Figure1, but the unbiased spectra used in subsequent calculations are estimated. The PSD of SSH is estimated by first removing the estimated constant error level to allow for an unbiased estimate of the spectral slope.

The authors have listened to your comments and have included a constant noise level and a slope fitted by the HY2B spectrum in Figure 1(If the spectra of each of the four satellites are plotted in the same figure by fitting straight lines to them, it may result in an image that is too cluttered and thus not conducive to clear observation and analysis.).

[Figure]

Figure 1 a. Along-track SSH averaged PSDs for HY2B, SARAL/ALTIKA, S3A, and SWOT within the Kuroshio Extension (**all spectra in Figure 1a are biased**, green arrows represent the range over

which PSD slopes were computed for conventional satellites, red arrows represent the range over which PSD slopes were computed for SWOT satellites, and the black solid lines show the spectral slopes which correspond to $k^{-5}$ and $k^{-11/3}$). b. The tracks of the first three satellites within the Kuroshio region distribution map. c. Distribution map of SWOT satellite tracks within the Kuroshio region (only two along-track data were selected for each pass). Red dashed lines represent the range over which the mean PSD was computed. **The horizontal four dashed lines represent the constant noise level for each satellite, while the two black dashed lines represent straight lines fitted to the unbiased spectra after removing constant noise from the HY2B and SWOT spectra, respectively.**

**Reference**

Chelton D B, Schlax M G, Samelson R M, et al., 2019. Prospects for future satellite estimation of small-scale variability of ocean surface velocity and vorticity[J]. PROGRESS IN OCEANOGRAPHY, 173: 256-350.

**Comment25**

l. 175: "…by removing the estimated constant error level below ..." Remove "below".

The authors have revised the reviewers' comments.

**Comment26**

l. 176-178: "Diverse methods for calculating the Power Spectral Density (PSD) […] in the estimated PSD slope range" : I agree but should not be here.

Thank you for the reviewer's comments. Following your comments, the authors have moved the discussion of "Diverse methods for calculating the Power Spectral Density (PSD) […] in the estimated PSD slope range" from lines 176-178 and adjusted it to lines 66-68 of the Introduction according to the nature of its content. The adjustments are as follows:

Dufau et al. (2016) proposed a method defined as the one-dimensional mesoscale resolution capability of altimetry satellites. The slope is determined by fitting the 90-280 km wavenumber spectrum and using 25 km below as the noise constant. **Notably, diverse methods for calculating the Power Spectral Density (PSD) can result in minor discrepancies in the slope range(Vergara et al., 2019). Additionally, data sampled at varying frequencies may also engender subtle variances in the estimated PSD slope range.**

**Comment27**

l. 179: "Hence, for the first three conventional missions, we chose wavelengths in the range of 70-250 km". Missing references: Dufau et al., 2016; Le Traon et al., 2008; Xu & Fu, 2011.

Thank you for pointing out the missing references in line 179, which we have added to the article.

**Comment28**

l. 184-185: "Due to the presence of […] etc (Boas et al., 2022)" : I believe this should not be a sentence on its own, but rather linked to the previous one for better coherence.

In response to your comments, we have merged the two sentences in lines 184-185 to enhance the coherence of the text. The revised sentence is as follows:

For the SWOT mission, the cross-correction process filters out some of the noise, which results in a continuous decrease in the spectral profile at less than 15 km (as shown in Fig. 1) , in addition to    many

sub-mesoscale phenomena such as internal waves and tides at 15-40 km (Boas et al., 2022).

**Comment29**

l. 185: "wavelength range" instead of "wavelength"

Thank you for your suggestion on line 185. We have amended the original text as you suggested, replacing "wavelength" with "wavelength range".

**Comment30**

l. 186-188: The one-dimensional mesoscale resolution capability essentially represents the shortest wavelength detectable in along-track altimeter observations where the signal exceeds the noise, correct? Perhaps this sentence could be rephrased for clarity.

Thank you for your suggestion regarding lines 186-188. You fully understand correctly that the one-dimensional mesoscale resolution capability is indeed the shortest wavelength that can be detected with signal strength exceeding the noise level in an along-track altimeter observation.

In order to improve the clarity of this expression, we have reworded the original text, and the revised sentence is as follows:

The intersection point where the error level and the spectral slope define the wavelength at which the PSD of the smallest-scale signal equals the error level. It is also called the satellite's one-dimensional mesoscale resolution capability.

**Global resolution capability of altimetry satellites**

**Comment1**

l. 207: "[...] theoretical predictions from SQG and GG theories." Add a reference please.

Regarding the "Theoretical Predictions of SQG and GG Theories", we have already supplemented the text with the following references: (Xu et al.,2011).

**Comment2**

l. 208-209: The authors should specify the figure number referenced in the cited paper.

The authors have taken your advice and have clearly labeled the figure numbers of the references cited in lines 208-209 of the article.

The manuscript was revised as follows.

**Figure 3 in the Dufau et al. (2016) article** pointed out that the calculated PSD shows an important energy peak near the 140 km wavelength at low latitudes.

**Comment3**

l. 212: "[…] oblique pressure tide […]" : Do the authors mean baroclinic instead of oblique?

Thank you for your careful review and valuable comments. Regarding the reference to "oblique" in line 212, what you have pointed out is indeed an important issue. After double-checking, we have confirmed that there was indeed an inappropriate use of vocabulary that led to a misunderstanding.

We have corrected the original manuscript and thank you again for your patience and professionalism.

**Comment4**

l. 216: "satellite altimetry observes" instead of "satellites altimetry observed"

Thank you for pointing out the vocabulary problem in line 216, which the author has corrected.

**Comment5**

l. 226: "for one month" instead of "for the one month".

Thank you for supplying your comments, the author has made changes to the original article.

**Comment6**

l. 227: "[…] results of previous studies." Add references please.

The author has taken your advice and has introduced a reference to the sentence. The reference is as follows: (Xu and Fu,2011; Dufau et al.,2016).

**Comment7**

l. 230: "northwest Pacific" instead of "Pacific northwest".

Thank you for pointing out the vocabulary problem in line 230, which the author has corrected.

**Comment8**

l. 239-241: "However, for the data from the period […] potential error in one of the satellite's corrections" Could the authors clarify this statement?

The satellite referred to in the paper is SARAL/Altika, and it is noted in the original paper that the trajectory effect of SARAL/Altika is more pronounced in the central Pacific and western Atlantic. We previously hypothesized that this might be due to a problem with one of the correction terms, but this is not accurate. In fact, the problem may lie in the orbiting accuracy of the satellite, or it may be related to the distance from the satellite to the sea surface (range) data collected by the laser ranging system. To further clarify the source of the problem, we designed and conducted the following experiments.

First, we plotted a scatter plot of SARAL/Altika along-track data in the Pacific Ocean (see Figure 5#). From Figure 5#, it can be seen that some of the tracks have obvious deviations in the longitude range of 160° to 190°. After further investigation, the files corresponding to these anomalous trajectories are::

- Altika_Dcycle175pass0468.txt
- Altika_Dcycle175pass0582.txt
- Altika_Dcycle175pass0784.txt
- Altika_Dcycle176pass0182.txt
- Altika_Dcycle176pass0496.txt
- Altika_Acycle176pass0231.txt

where "A" and "D" denote ascending and descending tracks, respectively, the number after "cycle" denotes the orbital period, and the number after "pass" denotes the specific track number. Figure 6# shows the spatial distribution of these anomalous tracks and the scatter plot of sea surface altitude anomalies (SLAs) with longitude. It is clear from the figure that the main problem is found in the descending tracks data, which is basically consistent with the trajectory effects illustrated in Figure 7#.

[Figure]

Figure 5# Scatter plot of SARAL/ALTIKA (longitude in horizontal coordinates, sla in vertical coordinates)

[Figure]

Figure 6# Trajectory Distribution Plot and Scatter Plot (In the Figure on the right, the horizontal coordinate is the longitude and the vertical coordinate is the value of sla)

[Figure]

Figure 7#Noise levels of SARAL/ALTIKA satellite

In order to further clarify which specific item in the satellite data is problematic, we take the Altika_Dcycle175pass0468.txt trajectory as an example and plot the SLA image of this trajectory after the correction term is processed, as well as the SLA image without the correction term (see Figure 7#). As can be seen in Figure 7#, whether or not the correction term is added only produces some minor differences in the SLA, but the data still shows a large deviation in geographic location. Therefore, the root of the problem may not be the correction term itself, but rather the bias in satellite orbiting accuracy.

This orbital deviation bias leads to a significant increase in the noise level of the data, which produces a noticeable trajectory effect in the error level plots.

[Figure]

Figure 8# The left graph shows the distribution of SLAs with the addition of the correction term and the right graph shows the distribution of SLAs without the correction term

In summary, the bias of trajectory effects in the SARAL/Altika satellite data mainly originates from the satellite orbiting accuracy problem rather than the correction term processing. The authors redescribe this component to enhance the academic rigor and traceability of the paper.

The manuscript was revised as follows (The bolded red font is the revised manuscript).

However, during the study period of this paper, the SARAL/Altika satellite's trajectory data in specific regions show significant deviations with a wide range of effects (see Figure .4b). Through detailed examination of the correction terms and comparative analysis of data with the same orbit number in two adjacent cycles, we find that the main source of this trajectory effect is the problem of satellite orbiting accuracy. The inaccuracy of the satellite orbiting leads to a significant increase in the data noise level, which results in a clear trajectory effect in the error level map.

**Comment9**

l. 243-244: How the authors explain the high-level noise pattern in the mid-north Atlantic for SWOT (Figure 3c)?

We thank the reviewers for their comments. The comment is highly scientific, which we did not fully consider in our previous analysis. To more clearly show the distribution characteristics of the high-level noise pattern, we have adjusted the color scale of Figure 3 in the manuscript and displayed the adjusted results in Figure 9#. By redrawing, we find that Figure 9# reveals not only the high noise level in the central Atlantic but also similar features in the west-central Indian Ocean. In order to further explore the causes of this phenomenon, we analyze the central North Atlantic as an example and try to reveal the potential reasons leading to the appearance of high noise levels.

[Figure]

Figure 9# Noise levels of SWOT satellite

In order to deeply explore the reason for the higher noise level, we calculated the power spectral density (PSD) of the sea surface altitude anomaly (SLA) along the track for the grid points located at (320°E, 10°N) and averaged them using the distance-weighted method, and the results are shown in Figure 10#. The right side of Figure 10# demonstrates the curve after weighted averaging, which has a slope of about 1.53, indicating that the PSD decays more slowly with frequency. This slow decay may result in high calculated noise levels in this region. In addition, Figure .11# shows the distribution of EKE in the mid-Atlantic. We find that regions with higher SWOT noise levels (e.g., Mid-Atlantic in Figure 10#) correspond to very large EKEs. We hypothesize that this large EKE may be a contributing factor to the high SWOT noise levels in this region. It should be emphasized that the results of the above analysis are still in the hypothesis stage and are only preliminary inferences based on current data. In the future, we will conduct a more in-depth study of the phenomenon of higher noise levels in the region in conjunction with data from more cycles, with a view to revealing the underlying causes.

[Figure]

Figure 9# Power Spectral Density Curve at Grid Points (320°E, 10°N)

[Figure]

Figure 11# Distribution of EKE for SWOT during the experimental period

**Comment10**

l. 256: Dufau et al. (2016) instead of Dufau C et al. (2016)".

Thank you for pointing out the error in the citation format of the original manuscript. The authors have corrected them in the revised edition.

**Comment11**

l. 269: "[…] where the peak slope occurs." Would it be possible to include a plot to visually represent this statement?

Thank you for your suggestion. Regarding the expression "[......]where the peak slope occurs", it has been visualized in Figure 1a in the original manuscript, with the exact location of the peak slope clearly marked. For better understanding, the authors have drawn a separate spectral map of the SWOT along-track data (Figure 12#). As can be seen from Figure 12#, the slope peak of the SWOT satellite is concentrated in the high wavenumber range (45-125 km) and has a low level of fitting error. As a result, the wavelengths corresponding to their intersections are also relatively short.

[Figure]

Figure 12# Along-track SSH averaged PSDs for SWOT within the Kuroshio Extension(blue curves are biased spectra, blue dashed lines are noise levels, orange curves are unbiased spectra, red dashed lines are the range of fitted slopes, and black dashed lines are the slopes fitted to unbiased spectra)

The manuscript was revised as follows

This is mainly due to SWOT's significantly lower noise level and the higher range of wavenumbers where the peak slope occurs **(Figure 1a).**

**Comment12**

l. 272: "Experiments have demonstrated…" Add references

The author has taken your advice and has introduced a reference to the sentence. The reference is as follows: (Fu et al., 2024).

**Global analyses of ocean-scale changes**

**Comment1**

l. 284: As I mentioned earlier, it would be better to refer to the cross spectral density.

The author has revised the original manuscript as you requested.

**Comment2**

l. 288: Marks et al. (2016) instead of Marks K M et al (2016).

Thanks to the reviewer for pointing this out. We have amended 'Marks K M et al (2016)' in line 55 to 'Marks et al. (2016)' to comply with formatting requirements.'

**Comment3**

l. 290: Why 0.5? Add some details please.

Marks et al. (2016) proposed that the coherence function can be used to assess the similarity between repetitive cycle signals and thus infer the resolving power. They defined the discriminability criterion for geodesic wavelengths as the spatial wavelength corresponding to a mean square coherence of 0.5. A similar study was carried out by Chelton et al. (2007), who defined the minimum scale at which a sea surface vortex can be resolved as the scale at which the ratio of the energy spectral density of the fusion product to that of the along-track data is 1/2. In signal processing, a mean-square agreement of 0.5 usually indicates that half of the power of two signals is correlated in the frequency domain. This not only implies that there is some correlation between the signals but also indicates that the signals are at this point at a comparable level to the noise. Therefore, when the coherence reaches 0.5, the corresponding wavelength can be regarded as an important threshold for the signal resolution capability.

Based on the above study, this paper also adopts 0.5 as the threshold value. Through cross spectral density analysis, we define the wavelength corresponding to a coherence of 0.5 as the critical value for ocean dynamics scale changes. This critical wavelength marks the occurrence of at least that scale of ocean dynamical processes, thus providing an important quantitative criterion for the in-depth study of ocean dynamical mechanisms.

The manuscript was revised as follows (The bolded red font is the revised manuscript).

Marks et al. (2016) proposed that the coherence function can be used to assess the similarity between repetitive cycle signals and thus infer the resolving power. They defined the discriminability criterion for geodesic wavelengths as the spatial wavelength corresponding to a mean square coherence of 0.5. Besides, a similar study was carried out by Chelton et al. (2007), who defined the minimum scale at which a sea surface vortex can be resolved as the scale at which the ratio of the energy spectral density of the fusion product to that of the along-track data is 1/2. In signal processing, a mean-square agreement of 0.5 usually indicates that half of the power of two signals is correlated in the frequency domain. This not only implies that there is some correlation between the signals but also indicates that the signals are at this point at a comparable level to the noise. Therefore, when the coherence reaches 0.5, the corresponding wavelength can be regarded as an important threshold for the signal resolution capability. In this study, we set the consistency threshold to 0.5, referring to standard practice in related fields.

Reference

Chelton, D. B., Schlax, M. G., Samelson, R. M., & de Szoeke, R. A. (2007). Global observations of large oceanic eddies. Geophysical Research Letters, 34(15).

**Comment4**

l. 294: Western Boundary Currents

The author has revised the original manuscript with your comments and thanks for your patience.

**Comment5**

l. 297: northwest Indian Ocean current

The author has revised the original manuscript with your comments.

**Comment6**

l. 306: "world's ocean current regions" Remove "world's"
The authors have taken your advice to remove redundant words from the original manuscript.

**Appendix B**

**Comment1**

l. 403-407: How are equations 5 and 6 linked? Is R the cross-correlation function and S the cross spectral density (Fourier transform of the cross-correlation)? Also, in equation 6, is P equivalent to S? Please be careful with the notation. Additionally, could you define all the variables?

Thank you for your interest in our paper and your comments. In response to your query on lines 403-407 about Eq. 5 and Eq. 6, it has been examined that there is indeed a sign inconsistency, where P and S are both power spectral densities. We have added this.
The manuscript was revised as follows.

We define the the self-coherence function of the random signal $x(t)$ is $R_x(\tau)$, the Fourier transform of $R_x(\tau)$ is defined as the self-power spectral density of $x(t)$ as shown in Equation (5). The self power spectrum density $S_x(f)$ contains all the information of $R_x(\tau)$,.

$$S_x(f) = \int_{-\infty}^{+\infty} R_x(\tau)e^{-j2\pi f\tau}d\tau \tag{5}$$

where $j$ is the imaginary unit, $f$ is the frequency, $\tau$ is the time delay. The mutual correlation characteristics of two random signals $x(t)$ and $y(t)$ can be described in the frequency domain by the cross power spectral density as shown in equation (6). The phase difference obtained by calculating the cross power spectral density can visualize the degree of similarity between the two signal sequences in the frequency domain(ZHOU et al.,2024).

$$S_{xy}(f) = \int_{-\infty}^{+\infty} R_{xy}(\tau)e^{-j2\pi f\tau}d\tau \tag{6}$$

The coherence function of the signals $x(t)$ and $y(t)$ for two repetition cycles is:

$$C_{xy}(\omega) = \frac{|S_{xy}(f)|^2}{S_x(f)S_y(f)} \tag{7}$$

Where $S_{xy}(f)$ is the cross spectral density of $x(t)$ and $y(t)$, and $S_x(f)$ and $S_y(f)$ are the self-power spectral densities of $x(t)$ and $y(t)$ respectively.

**Comment2**

l. 408-409: "The coherence function can judge the […] cycle signals to infer resolution capability." Already mentioned earlier, see l. 397-398. Marks et al. (2016) instead of Marks K M et al (2016).

Thanks to the reviewers for their careful review. We have deleted redundant sentences from the text and rechecked and corrected the citation format of the references to ensure accuracy and consistency.

**Comment3**

l. 417: Which kind of interpolation is performed? Linear?

Thank you for your question. In our study, we did use the linear interpolation method. The reason we chose linear interpolation is that it provides a simple and effective solution in dealing with data, and it is especially suitable for situations where the changes between data points are relatively smooth. In addition,

linear interpolation has a low computational complexity and meat the requirement of the study.

The original manuscript has been supplemented as follows:

Since the along-track data of the two repetition cycles required strict alignment, **linear** interpolation was required to fill in the missing data.

**Figures**

**Comment1**

Figure 1: The authors should specify whether these spectra are biased or unbiased.

Thanks to the reviewers for their comments! Regarding the curve in Figure 1a, although the spectra shown in the figure are biased; in calculating the slopes we use the processed unbiased spectra. The noise level is removed from the power spectral density (PSD), which enables an unbiased estimate of the spectral slope.

To present the authors' experimental procedure more clearly, the authors have made the following changes to line 175 of the original manuscript and have added a note to the figure notes of Figure 1.

175:The PSD of SSH is estimated by first removing the estimated constant error level to allow for an **unbiased estimate** of the spectral slope.

Note to Figure 1:Figure 1 a. Along-track SSH averaged PSDs for HY2B, SARAL/ALTIKA, S3A, and SWOT within the Kuroshio Extension (**all spectra in Figure 1a are biased**, green arrows represent the range over which PSD slopes were computed for conventional satellites, red arrows represent the range over which PSD slopes were computed for SWOT satellites, and the black solid lines show the spectral slopes which correspond to $k^{-5}$ and $k^{-11/3}$).

**Comment2**

Figures 2, 3 and 5c: The units are missing.

Thanks to the reviewers, we have added units to the figure notes for Figures 2 and 3. For Figure 5c, the author has re-added unit to the colorbar. The revised manuscript is as follows (the numbering of the images has changed due to the addition of images in the original manuscript):

**Figures 2**:Figure 3 Slope maps for different satellites ((a) HY2B, (b) SARAL/ALTIKA, (c) S3A, (d) SWOT. **Units=log(m²/cpkm)/log(cpkm)**)

**Figures 3**:Figure 4 Noise levels **(m rms)** of different satellites ((a) HY2B, (b) SARAL/ALTIKA, (c) S3A, (d) SWOT)

**Figures 5c:**

[Figure]

Figure 6 Resolved wavelengths for different satellites (a. HY2B, b. SARAL/ALTIKA, c. S3A, d. SWOT)

**Comment3**

Figure A1: The legend mentions some orange points, but I cannot see them in the plot.

Thanks to the reviewer for pointing out the problem in Figure A1. It was my inattention that led to the misrepresentation. I will correct the title of Figure A1 in the revised version to ensure that the legend is consistent with the content of the figure.

The manuscript was revised as follows.

[Figure]

Figure A1 Schematic diagram of the satellite trajectory and the grid points to be solved, where the blue points are the grid points to be solved, the yellow points are the other grid points, t**he black dashed lines** are the trajectories of the satellites, and the **red lines** represent the range of the box**.**

**References**

**Comment1**

l. 481 and 499: The reference formatting is inconsistent.

Thanks to the reviewers for pointing out the inconsistency in the formatting of the references in lines 481 and 499. The authors have made the corrections as follows:

Tchilibou, M., Gourdeau, L., Morrow, R., Serazin, G., Djath, B., & Lyard, F. (2018). Spectral signatures of the tropical Pacific dynamics from model and altimetry: a focus on the meso-/submesoscale range. Ocean Science, 14(5), 1283-1301.

ZHOU, R. S., ZHANG, S. J., & KONG, X. X. Investigation on Along-Track Geoid Resolution Capabilities of HY-2 Based on Spectrum Analysis. Journal of Northeastern University (Natural Science), 44(9), 1328.

---

## Author Response (AR2)

**Editor decision**

Thanks for your uploaded manuscript and thanks for addressing the reviewers' comments which were very thorough. However, I would like to suggest some changes to the caption of some of your figures in the Appendix which I think will add clarity to your article, particularly that of Fig A2 and clarify a bit more the concept of the "share of the different distances".

Dear Editor:

The author's team would like to thank you for reviewing the paper and providing useful feedback and suggestions. We have carefully read and responded to your comment. Your comment is in black font, our explanatory response is in blue font, and the corresponding revision in the manuscript is in red font.

**Problems with the title description of Figure A2.**

Thank you for your valuable comments. The lack of clarity in the caption you pointed out does exist, which may cause inconvenience to readers in understanding the content of the Figure. I have therefore reworked the title of Figure A2 to more clearly convey the approach of the paper.

We calculated the distance (unit: °) between the data along the track and the reference point and used the Inverse Distance Weighting (IDW) method to assign weights. Since the final results are presented as a $2° \times 2°$ grid, we segment the distance between the orbit and the reference point as follows:

➢ When the distance is less than or equal to 1°, it is uniformly calculated as 1°;

➢ When the distance is greater than 1° and less than or equal to 2°, it is uniformly calculated as 2°;

➢ When the distance is greater than 2° and less than or equal to 3°, it is uniformly calculated as 3°;

➢ When the distance is greater than 3° and less than or equal to 4°, it is uniformly calculated as 4°;

➢ When the distance is greater than 4°, it is uniformly calculated as 5°.

The manuscript was revised as follows:

[Figure]

Figure A2 shows a schematic diagram illustrating the partitioning of distance between the data along the track and the reference point. (The green area indicates that the distance between the orbit and the reference point is less than or equal to 1° and is calculated uniformly as 1°, the pink area indicates uniformly as 2°, the orange area indicates uniformly as 3°, the brown area indicates uniformly as 4°, and the black area indicates uniformly as 5°)